# Block Caving Mining Method: Transformation and Its Potency in Indonesia

**Sari Melati** [1,2,3], **Ridho Kresna Wattimena** [1,3], **David Prambudi Sahara** [4,5,*], **Syafrizal** [3,6], **Ganda Marihot Simangunsong** [1,3], **Wahyu Hidayat** [5,7], **Erwin Riyanto** [5,8] **and Raden Roro Shinta Felisia** [9]

1   Geomechanics Research Group, Faculty of Mining and Petroleum Engineering, Institut Teknologi Bandung, Bandung 40132, Indonesia
2   Mining Engineering Study Program, Faculty of Engineering, Universitas Lambung Mangkurat, Banjarbaru 70714, Indonesia
3   Mining Engineering Study Program, Faculty of Mining and Petroleum Engineering, Institut Teknologi Bandung, Bandung 40132, Indonesia
4   Global Geophysics Research Group, Faculty of Mining and Petroleum Engineering, Institut Teknologi Bandung, Bandung 40132, Indonesia
5   Geophysical Engineering Study Program, Faculty of Mining and Petroleum Engineering, Institut Teknologi Bandung, Bandung 40132, Indonesia
6   Earth Resources Exploration Research Group, Faculty of Mining and Petroleum Engineering, Institut Teknologi Bandung, Bandung 40132, Indonesia
7   Geophysical Engineering Study Program, Faculty of Technology Mineral, UPN Veteran, Yogyakarta 55283, Indonesia
8   Geoengineering Division, PT Freeport Indonesia, Mimika 99968, Indonesia
9   English Literature Study Program, Faculty of Letters and Cultures, Universitas Gunadarma, Jakarta 16452, Indonesia
*   Correspondence: david.sahara@gf.itb.ac.id

**Abstract:** The block caving mining method has become increasingly popular in the last two decades. Meanwhile, Indonesia has several potential ore bodies which have not yet determined suitable mining methods. The references to block caving mining projects worldwide and the potency of metal deposits in Indonesia were reviewed to determine the requirements of ore bodies suitable for mining using the transformed block caving method. This method can be applied on a blocky ore body with a thickness of 200–800 m, various rock mass strengths until 300 MPa, from low to high (from 0.3% Cu until more than 1.0% Cu), but of uniform grade and at a depth from 500 to 2200 m. The technical specifications for running block caving mines have been synthesized, including preparation methods, undercutting strategy, mine design, mining equipment and monitoring. Considering the requirements and the successful practice of the block caving project in the Grasberg Caving Complex as a role model, the Indonesian government should concentrate on the detailed exploration of porphyry deposits and feasibility studies on applying the method to the prospective ore bodies, i.e., Onto, Tambulilato, Tumpangpitu and Randu Kuning. In addition, the exploration method, cost, operation, environment, mining policy and social geology are important aspects worth noting.

**Keywords:** caving; Indonesia; mining; porphyry; underground

## 1. Introduction

Mining production has increased significantly in the last two decades, especially driven by technological mastery in mineral exploration, mining and processing. Mining escalation started in 2002 after digitalization and communication rapidly spread worldwide. Excluding the slight decline in mining production in 2009 and 2016 due to the global crisis, the general trend in mining production is increasing with an average annual increase of 20.7% [1]. This was mainly because of the high demand for raw materials for electronic devices, communications and interconnection networks. Therefore, investment in the

search for new resources and reserves, as well as replenishment in funding for upgrading productivity, is becoming more necessary than ever. This momentum contributes to advancements in the mining sector in general, including improving the reliability and efficiency of mining methods.

The exploration of safe and clean mining on Earth and asteroids has become an important issue lately, especially in preventing safety accidents and environmental damage. Micro-seismic monitoring, along with stress and blast vibration monitoring, is a reliable technology both in open pit and underground mining for early detection of safety hazards and building an effective emergency rescue system [2]. One of the important achievements in micro-seismic monitoring is that the dynamic fracture formation process around the longwall mine can be explained by clustering methodology. The accurate cluster location can be determined by dividing a continuous group of mining seismic events and relating them to some parts of the rock mass [3]. These technological developments in mining practice are expected to increase the opportunities for mining sustainability. The combined solution of the Triple Helix Model (THM) between government, industry and university, as well as the Open innovation (OI) concept and Environmental, Social, and Governance (ESG), should enable sustainable development in a specific country and globally [4].

Most of the highly productive mines in the world are excavated on the surface. Surface mining is more profitable as it has exceptional advantages in flexibility and mobilizing. Therefore, realization design for producing as much ore as possible is convenient. Unfortunately, surface mining has an economic limit when the remaining ore reserve becomes deeper. In that case, underground mining methods are the only option. In addition, underground mining is assumed to leave fewer environmental impacts than surface mining [5]. Among underground mining methods, block caving is the most cost-effective as it can produce 10,000–100,000 tons per day with a relative operating cost of USD 1 to 2.5 per ton [6,7]. In the last twenty years, existing block caving mines have varying ore body thickness ranges from 200 to 800 m. A study proposes that the cut and fills stopping method was the optimal underground mining method for deep mining (>800 m below the ground surface), compared with block caving and four other methods [8]. However, it is only relevant for the lead-zinc-silver Trepca mineralization deposit investigated in the study, which has an irregular shape and ore thickness of 30–100 m. Other studies on underground coal mining noted that non-pillar mining or Longwall Mining, a caving method for coal deposits, had the lowest environmental burden and was determined to be the optimal mining method [9]. Thus, the caving method can be favored as an underground mining method due to its high productivity and low environmental impact. The deposit size must be huge enough to justify the investment costs at the beginning of production.

In the caving mining method, mining is achieved by breaking most or all of the ore body. Caving mines are classified as Longwall, Sublevel Caving and Block Caving by requirement factors, including ore strength, rock strength, deposit shape, deposit dip, deposit size, ore grade, ore uniformity and depth. Longwall is recommended for any ore strength, weak/moderate rock strength, tabular deposit shape, low/flat deposit dip, thin/wide deposit size, moderate ore grade, uniform grade and moderate/deep depth [10]. Due to coal seams having a relatively flat dip, the longwall method is usually applied in coal mining. Longwall mining in coal seams is an underground mining technique where a tabular block longwall panel of coal with a typical length of 1.5–3.0 km, a typical width of 200–300 m and a typical height of 3.0–4.5 m is extracted. Two pairs of roadways are first driven outside the panel within the seam for access. Machines used for operations are drum shearer machine as coal cutter, belt conveyor for hauling and hydraulic-powered roof supports providing temporary support during coal cutting [11]. Longwall mining is applied intensively in Australia, China and America and also in Ukraine, India, Turkey, Bangladesh and Poland, [11–28]. In Indonesia, the longwall mining method is applied in Kutai Kertanegara, East Kalimantan, at the mining concession of PT Gerbang Daya Mandiri (GDM). GDM recoverable sub-bituminous coal reserves are approximately 29.2 million tons and one million tons of annual production have been planned [29–33].

The second caving method, Sublevel Caving, is an underground mining method proposed for moderate and strong strength of ore, weak rock strength, tabular/massive deposit shape, steep deposit dip, large thick deposit size, moderate ore grade, moderate uniformity grade and moderate depth [10]. Sublevel caving is a mass mining method in which the ore is drilled and blasted while the waste rock caves and fills the space created by the extraction of ore. The ore body is divided into vertical intervals called sublevel intervals. The ore within each sublevel interval is drilled in a fan-shaped design at a constant horizontal distance along the production drift. Load Haul Dump machines load muck pile from the draw point [34]. Dilution becomes the issue in this method due to the strength of the ore body and rock mass. Sublevel caving is applied in iron mines in Ukraine, iron mines in Sweden, iron oxide mines in Norway, coal mines in Spain, coal mines in India and gold mines in Australia [34–42].

The last caving method, block caving, is recommended for moderate and weak ore and rock strength, weak rock strength, tabular/thick deposit shape, steep deposit dip, very thick deposit size, moderate ore grade, moderate uniformity grade and moderate depth [10]. The first documented underground mine which applied the block caving method was the Pewabic iron mine in 1895 [43]. Although the block caving method has been known for over a decade, its massive deployment has only occurred in the last twenty years. During that time, the number of cave mining projects increased almost four times, from 17 to over 50 [44,45]. Not only in number, but caving mines were also getting larger and deeper. In the beginning, footprint areas and block heights were less than 100,000 $m^2$ and 200 m, respectively, with the maximal overburden depth of 600 to 700 m. Recent cave mines have footprint areas greater than 400,000 $m^2$ and block heights exceeding 400 m, with an overburden depth of up to 1200 m [46,47]. The thickness of the ore body mined using the block caving method varies between 200 and 800 m. As an indication, the ore body widths of Northparkes, Ridgeway, Stornoway, Palabora, Grasberg, Oyu Tulgui and El-Teniente are 200 m, 200 m, 225 m, 250 m, 400 m, 500 m and 500–800 m, respectively [47–57].

The most significant evolution in the block caving method is related to the strength of the caved rock mass. Regarding ore and rock strength criteria, block caving was designed for weak or moderate rocks [10]. The International Society of Rock Mechanics (ISRM) Commission on the Classification of Rocks and Rock Masses in 1981 classified rock mass with uniaxial compressive strength of 6–60 MPa as suitable for block caving. Thanks to massive underground mining technology innovation, block cave is now implemented in competent rock; for instance, rock mass with the highest UCS (138 MPa) in Cadia East, 144 MPa in Northparkes, 157 MPa in Deep Mill Level Zone, 170 MPa in El Teniente and 300 MPa in Palabora [58–60]. This condition is classified as a high-strength rock. Some blocks mentioned above in caving fields took advantage of rock mass preconditioning to accelerate production, as well as to increase the safety of the workers. Rock mass preconditioning aims to generate new cracks or to elongate the extent of in-situ cracks using hydraulic fracturing and/or destress blasting on the competent rock. Some block caving in competent rock mass performed the rock preconditioning during the preparation and the development zone before the regular block caving processes [58–60].

All these improvements have enabled the block caving method to majorly contribute to the supply of ores from underground mines. Five of the ten underground ore mines with the highest production levels in 2020–2021 use the megatons' block caving method [61,62]. These are copper, gold and silver mines at Grasberg Operations (PT Freeport, Indonesia), Cadia Valley (Newcrest Mining, Australia), Padcal (Philex Mining, Philippines) and New Afton (New Gold, BC, Canada). In addition, diamond mines are located at Udachy (Alrosa, Russia), where the total ore processed per year is between 3.39–51.53 million tons.

In Indonesia, PT Freeport Indonesia introduced the block caving method. There are four of five ore bodies in the PT Freeport Indonesia underground complex, which are mined by the block caving method, i.e., Deep Ore Zone (DOZ), Deep Mill Level Zone (DMLZ), Grasberg Block Cave (GBC) and Kucing Liar (KL). This method is designed to produce 20,000 to 130,000 tons of ore per day. Grasberg caving complex is categorized as super

caves with Chuquicamata and New Mining Levels at the El Teniente project in Chile, Oyu Tolgoi projects in Mongolia and the Resolution Copper project in Arizona [63].

Located in an active tectonic region, Indonesia is famous for hosting large rock mass intrusion bodies suited to the block caving method. However, only a few have been operated because the application takes time for technology readiness, feasibility studies, and development. As a first step, the potential deposits must be explored in detail. Since the block caving mining practice plays an essential role in maintaining or even upgrading the total national capacity of rock mass mining in Indonesia, the development and potential growth of the block caving method application in Indonesia have been reviewed. The discussions on the transformation and potency of block caving in Indonesia are based on four aspects, i.e., technological advancement in block caving, geologic and tectonics analysis in Indonesia, existing operations, and non-technical aspects. It is expected that the opportunities to spread the block caving method in mineral exploitation, especially in Indonesia, can fulfill the growth in energy demand faced today and in the following decades. This study is important in dissemination for practitioners, engineers and academics regarding the block caving method, which is the most efficient method of modern and future mining for huge ore bodies. Studies in Indonesia are necessary to provide recommendations for the potential application of this mining method in utilizing the country's natural resources (especially metal deposits).

## 2. Block Caving Method

### 2.1. Initial Block Caving for Weak to Moderate Rock

Block caving becomes attractive when deposits near the surface appropriate for the open pit mining method are harder to find. Furthermore, open pit mining has an economical depth limit of the ore body that can be extracted. Because of this limitation, ore bodies with low-grade levels at a greater depth cannot be mined using an open pit. On the contrary, block caving has relatively high productivity and inexpensive production costs, but with a higher risk in terms of technology and safety.

In the underground mining classification system, block caving is a method for mining weak to moderate rock and ore (Table 1). Block caving can extract rock on a large scale following the geometry of the cave propagation. As the cave propagates, ore mass at the top of the cave and the edge of the abutment will be heavily fragmented and eventually free fall to be excavated at the production level. As block caving can excavate large rock mass, it can be applied to almost all rock grades, low to high. The method is applied to mineral deposits such as iron ore, copper, molybdenum mineralization and diamond-bearing kimberlite pipes [64]. Recently, it has been used: on porphyry deposits, kimberlite deposits, skarn and porphyry-related deposits, asbestos mines, iron ore deposits, sedimentary exhalative (sedex) deposits, volcanogenic massive sulfide (VMS), stratiform deposits and others [65].

Figure 1 shows the general description of the block caving method. The accumulation of stresses at the top of the cave due to gravity and internal stress is enough to frack and break rock mass naturally if some conditions are fulfilled, e.g., regarding the radius of the cave and rock strength. The first stage is the production or extraction level development below the ore body. The undercut level was commonly excavated about 10–20 m above the production level, depending on the thickness of the overburdened rock and in-situ stress conditions. Pillars between the undercut drifts are then drilled and blasted to build slots below the orebody. This activity is known as undercutting. Draw bells are constructed between the production and undercut levels to accumulate the broken ore that had fallen from the top of the cave. Sometimes, secondary fragmentation using a jackhammer or secondary blasting is required if the block size of broken ore is larger than the requirement of the processing stages. Well fragmented ore is transferred to draw points at the production level through the ore pass.

**Table 1.** Underground Mining Classification Characteristic (Hartman, *Introductory Mining Engineering*, 1987).

| Underground Method | Unsupported | | | | Supported | | Caving | | |
|---|---|---|---|---|---|---|---|---|---|
| Factors | Shrinkage stoping | Sublevel stoping | Stope and pillar | Room and pillar | Cut and fill stoping | Square set stoping | Longwall | Sublevel caving | Block caving |
| Ore strength | Strong | Moderate/strong | Moderate/strong | Moderate/strong | Moderate/strong | Weak | Any | Moderate/strong | Weak/moderate |
| Rock strength | Strong | Fairly strong | Moderate/strong | Moderate/strong | Weak | Weak | Weak/moderate | Weak | Weak/moderate |
| Deposit shape | Tabular/lenticular | Tabular/lenticular | Tabular/lenticular | Tabular | Tabular/lenticular | Any | Tabular | Tabular/massive | Tabular/thick |
| Deposit dip | Fairly steep | Fairly steep | Low/moderate | Low/flat | Fairly steep | Any | Low/flat | Fairly steep | Fairly steep |
| Deposit size | Thin/moderate | Thick/moderate | Any | Large/thin | Thin/moderate | Usually, small | Thin/wide | Large thick | Very thick |
| Ore grade | Fairly high | Moderate | Low/moderate | Moderate | Fairly high | High | Moderate | Moderate | Low |
| Ore uniformity | Uniform | Uniform | Variable | Uniform | Variable | Variable | Uniform | Moderate | Uniform |
| Depth | Shallow/moderate | Moderate | Shallow/moderate | Shallow/moderate | Moderate/deep | Deep | Moderate/deep | Moderate | Moderate |

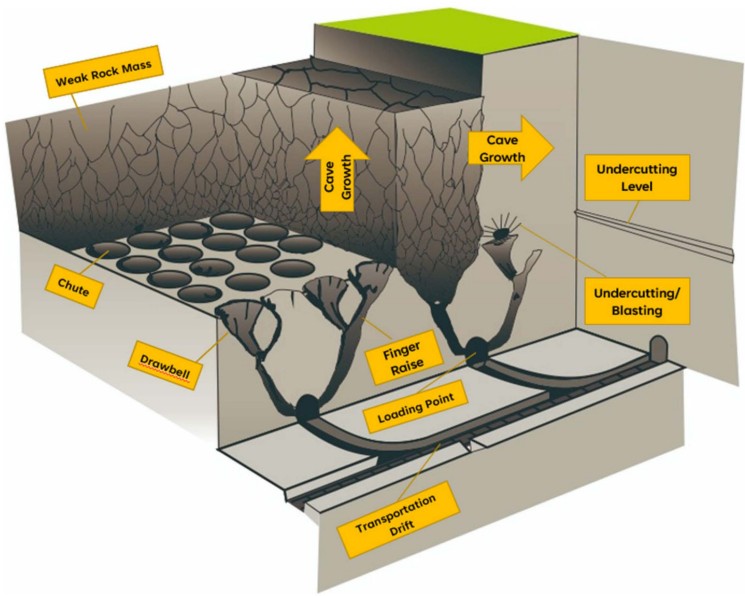

**Figure 1.** Layout of initial block caving. Undercutting blasting is generated to facilitate cave growth through weak rock. Draw bells and Finger raises are used to transport the falling ore to the loading point. The ore is transported to the transportation drift.

The drawing of blasted ore induces the flow of caving material and removes the support of the cave back. The cave back fails, and the muck pile fills the space formed by undercutting activity. The cave will continue to propagate upward if the ratio between the area and the circumference of the cave has satisfied the in-situ rock strength. In this case, blasted ore falls to draw points at the production level. This continuous caving process is the expected flow production in the block caving method. The last mining cycle is loading and hauling from the production level to the processing facility on the surface. The footprints of cave mining are usually built in several thousands of square meters. Typically, the development takes about ten years from the first access drifting until the first production when ore mass reaches the cave propagation and collapses.

In the early underground mining references, e.g., Pewabic iron mine, the block caving mining method was suited to ore bodies with the characteristics below [10,66].

1. A weak ore body can easily be fractured or fail and be separated around the block.

2. A weak wall rock breaks into bigger boulders than an ore fragment, where the pressure helps to break the ore body below.
3. Homogeneous deposit shape is required, as it is impossible to conduct selective mining. Should eye catching characteristics cause physical differentiation between ore body and capping, dilution at the draw point can be avoided. Ore body should be difficult to react with air. Therefore, this method is not appropriate for sulfidation deposits.
4. Dip of the deposit is not a problem. However, a dip > 65° is favorable if it is a vein.
5. Deposit thickness > 3 m with height > 35 m.
6. The grade of ore should not be high.
7. The depth is moderate.

The grade of ore should not be high (point 6) as the uniformity of grades in the ore body is more important than a certain percentage of grades, because in this method almost all parts of deposits can be recovered. Considerations of mine feasibility depend more on the volume of ore, along with the other contained minerals, that will determine the life of the mine. The high-grade Cu deposits containing low-grade other minerals is equivalent with the low-grade Cu deposits containing high-grade other minerals. As quantitative values for comparison, Ridgeway Deeps has 0.38% Cu and 1.80 g/t Au [49–51], while El-Teniente ore body has 0.62–0.98% Cu, 0.019% Mo, 0.005 g/t Au and 0.5 g/t Ag [55,66]. Both are mined economically using the block caving method.

Later, with the more quantitative approach, Miller-Tait, 1995 [67] stated that the block caving method is most appropriately applied to mine deposits with these characteristics:

1. A massive ore body with a thickness of more than 100 m, a dip of more than 55° and depth of more than 100 m.
2. Grade distribution is relatively uniform.
3. Very low-quality ore body (Rock Mass Rating, RMR = 0–20), wall rock is from very weak to moderate (RMR = 0–60).
4. The ore body and wall rock's uniaxial compressive strength ($\sigma_c$) are very weak. Compared with major principal stress ($\sigma_1$), the ratio $\sigma_c / \sigma_1$ is lower than 5.

### 2.2. Transformation to Competent Rock

To accelerate the technology and safety of block caving, caving practicians routinely discuss the current block caving technology, paradigm-shifting and recent developments in block caving at the International Conference on Block and Sublevel Caving. These events were held five times in Cape Town (2007), Perth (2010), Santiago (2014), Vancouver (2018) and Adelaide (2022). Together with other cave mining methods, panel caving and sublevel caving, block caving transforms itself in order to improve its viability, safety, cost, production and profitability. Research and practice were performed to develop solutions that reduce lead times and capital investment for the near future.

Industry research and innovation mostly contribute to upgrading the implementation of recent technology in block caving practice and benchmarking. The experiences of mining consultants and contractors in servicing different sites enrich a mature and flexible operation. University or research institutes support a profound understanding of rock behavior and its response during mining, develop modified formulation or classification systems and deploy new approaches to solve the latest problems. Collaborative research between all interested parties is a valuable process.

The newest update in block caving transformation announced in 2022 is the modified block caving method called Raise Caving. The fundamental difference with the conventional block caving method is that this system uses a hoisting system for mucking the ore through the raise, which is developed vertically at the center of the ore body from the top to the bottom. Boring, charging of explosives and supporting the raise are conducted on the platform placed in the hoisting system. This platform is removed from the raise during blasting activity and placed at the station's level to avoid damage [68–72]. The method consists of two phases, the de-stressing phase, and the production phase. The de-stressing

stages consist of the slot rise, the start slots and the slots that are developed from top to bottom of the ore body. In the production phase, de-stressed rock masses are blasted and fall to the draw points. The caving rises from bottom to top. Ladinig described raising caving as a hybrid method starting from a pillar-supported method in the early phase, converting to an artificially supported shrinkage-stopping method during the production phase and ending up in a caving method as stopes are drawn empty. Raise caving was successfully tested for a pre-feasibility study on Kiruna Mine. It reduces approximately 50% of the cost of infrastructure development compared with the conventional caving method [71].

In the last two decades, block caving underwent a series of transformations in terms of mining equipment and technologies. It impacted the escalation of production rates and scale. At the start of the block cave mine in 1898, the production rate was about 2–8 kilo tons per day as it was only implemented on a weak rock at shallow depths. There was no significant improvement for 80 years, during which production rates increased steadily to 10–20 kilo tons per day. The mechanized system began to be utilized in the 1970s. All active cave mines nowadays use Load Haul Dump (LHD) for material handling that could achieve 20–40 kilotons per day. LHD is like conventional loaders but was developed for the toughest rock mining applications, taking overall production economy, safety and reliability into consideration. In addition, induced micro-seismicity due to mineral extraction conducted in deep underground mining is continuously monitored to minimize the seismic hazard. Thanks to the improvement in block caving technology in the last decade, the production capacity increased to 80–160 kilotons per day.

Technical improvements in mining extraction and safety have allowed block caving to be applied on moderate to strong rock [73–78]. In some block caving, hydraulic fracturing, and or destress blasting were implemented before the undercutting process started [58,60]. Hydraulic fracturing aims to decrease rock mass strength by stimulating new fractures and extending existing fractures. These preparation stages are called preconditioning. Figure 2 shows the scheme of block caving on a large scale, where preconditioning rock mass is conducted on the next excavating process target.

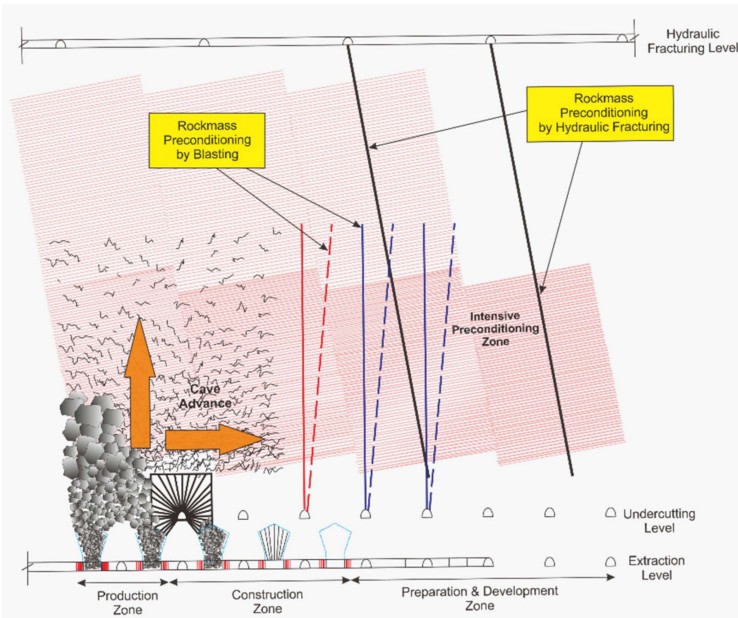

**Figure 2.** Layout of modern block caving with hydrofracturing and destress blasting for preconditioning the competent rock mass. In this example, rock mass preconditioning by blasting is carried out through a well drilled from the undercutting level (blue and red lines) and hydraulic fracturing through a well drilled from hydraulic fracturing above the cave. The falling material from the propagated cave is mucked from the extraction level.

The layout of intensive preconditioning, where hydraulic fracturing and de-stress blasting were combined, is also shown in Figure 2. These techniques enabled competent rock to be mined using block caving because of rock fragmentation before production. The hydraulic radius of the block caving is also increased due to the higher rock strength of the competent rock. For instance, in a weak rock required hydraulic radius is only a maximum of 50 m. However, now block caving mines in competent rock reach continuous caving at minimum hydraulic radius of 60–70 m and an undercutting area of around 280 m × 280 m.

In mining production technology, autonomous LHD is an excellent technology that has been relied on for the prodigious rate of major block caves productivity. This improves mining safety and, therefore, enables block caving in low-grade and larger deposits. A tele-remote system was experimented with in the 1980s with the main goal of removing the operator from the mucking process at the draw points. The recent version of autonomous LHD enables one operator to control 2–3 LHDs from a central control station. Henderson mine, Colorado, Magma mine, Arizona, and the Andes, Chuquicamata, Chile, Grasberg, Indonesia and Oyu Tulgui, Mongolia are cave projects that have been utilizing this autonomous system [79].

In terms of modeling, numerical modeling offers a fast and representative rock mechanics assessment when several designs need to be analyzed. Numerical modeling was used for the first time in 1973 to study cave propagation behavior at the El Teniente Mine in Chile [80]. In the beginning, numerical modeling of block caving is limited to two-dimensional, elastic behavior and a continuum model. Along with developing tools and applications, it upgrades to the three-dimensional, elastoplastic behavior, and discontinuous or hybrid model. Most of the numerical modeling is exploited to simulate the stress–strain condition, rock mass behavior around the cave, cave propagation mechanism, and subsidence behavior, or compared to the other results of empirical, physical, or analytical methods. Discontinuous and hybrid models are developed specially to accommodate the modeling of heterogeneous or defected rock and its flow in the caving process [81,82]. Cellular automata, an artificial intelligence with iterative, stochastic, and discrete mathematical models coupled with its predecessor numerical tools, is the recommended approach for block caving modeling. Caving mechanics models were then utilized for production scheduling [83–88].

Mining companies in the USA and Australia, Freeport McMoran, Newcrest Mining and Rio Tinto, are the leading companies in block caving mining practice. Their success in operating several sites becomes a reference for other mining companies in practicing this method. They noted that the geotechnical setting is always a determinant factor of mine planning and design. Freeport McMoran successfully redesigned the mine and modified ground support in long- and short-term planning at Grasberg Block Caving when faced with heterogeneous rock masses [56]. Newcrest Mining considered fragmentation, hydraulic fracturing, mine design, cave management and the stress abutment zone as geotechnical considerations for mine planning and design in the Cadia Valley Operation [49]. Rio Tinto, with its recent experiences in Grasberg, Palabora, Northparkes, Argyle and present projects in Oyo Tolgui and Resolution Mine, but cautioned that detailed measurement and monitoring are the most important steps in the design, construction and operation of block caving mines [57].

Productivity, grade and mining value are sensitive aspects in operating block caving projects [89]. As the most promising advanced underground mining method after open pit closure, it is necessary to consider whether the exposure on the surface and the risk of slope failure of the open pit affect the grade of the underground ore body [90]. Regarding this issue, the concept of "Cave to Mill" was introduced to provide consistent tonnage and grade feed which consists of (i) better characterization of the material reporting to draw points, (ii) measurement of the variation in metal content that is delivered from draw points and (iii) a bulk sorting system which offers flexibility and control [91]. Sensor-based sorting applications and automation of an online grade analyzer are helpful tools for intensifying the capacity of material handling and the efficiency of the quality control process [89–91].

Understanding the actual condition of rock mass is vital in mine stability analysis. In terms of actualization aspects, monitoring is the best method for quantifying rock mass and is used as a reference for validating and verifying predictive models. Micro-seismic monitoring is also the most effective method for block caving compared to other monitoring methods (displacement monitoring with dilatometer, multiple borehole extensometer, borehole camera, scanner, radar, or aerial photographs). This condition is due to its capacity, which covers a large area and records the spatio-temporal alteration. In addition to being sensitive to movement, the data obtained is very comprehensive and flexible for various advanced analysis purposes. Micro-seismic monitoring methods have been applied in several caving mines, using both passive and active seismic sources. Based on our previous study, the placement of the seismometer network significantly affects the resolution and level of data certainty [92]. The collaboration of analytical methods and iterative solutions has helped to localize the source of seismic events accurately. The accuracy of the location of the source of this event is essential for modeling geological structures in rock mass [93].

## 3. Requirements and Technical Specifications for Block Cave Mines Globally

Currently, there are over 50 cave mining projects in various stages of study and development in the world (Figure 3). They have spread mostly in North America, Australia, Africa, and South America. Some valuable projects are also located in Asia, i.e., in Indonesia, the Philippines and Mongolia. The several specifications of the existing block caving mines in the world are reported in Appendix A, Table A1. The block caving method has transformed from the original limited version to the recent version that is more productive, more efficient, with higher technology, and safer. The block caving method in the future may become competitive in terms of its development process if it can keep following the current rapidly evolving trend, be more flexible in applying varied depths and grades of the ore body and maintain minimum risks associated with mining hazards. Based on the best practice and available references for block caving [94], e.g., Palabora and Cullinan in South Africa; Grasberg Caving Complex in Indonesia; Northparkes Mine, Argyle, Ridgeway Deeps, Cadia East and Carrapateena in Australia; Andina and El Teniente in Chile; Renard and Red Chris in Canada; Resolution Mine in Arizona; Salvador in Central America; Jwaneng in Botswana; Padcal in the Philippines; and Oyu Tolgoi in Mongolia, block cave mines' requirements and technical specifications could be summarized as follows.

Mine planning becomes crucial to achieve the best tactical short-term and long-term strategies, since it ensures the life of the robust block caving on strong rock masses. In the development stage of block caving, especially if using autonomous hauling, the haul distance from the production level at the bottom of the ore body to the processing plant on the surface must be designed as efficiently as possible to ensure the smooth flow of transportation equipment. In addition, structural design, functional design and maintenance management system design should deal with realistic and appropriate design limits [95,96].

In the undercutting stage, the optimization of the undercut is the main parameter in design. It imposes stress perturbation, hydraulic radius, cave propagation, caving direction and the stability of pillars at the extraction level [97–99]. The main target of undercutting competent rock mass is to achieve a large hydraulic radius while keeping the level stable. To do so, the height of the undercut cannot be less than 10 m and requires wider pillars and powerful rock support on the extraction level. This mining process makes the block caving work like an underground rock factory [99].

Block caving has several risks related to rock mass instability. Rock burst and strain burst are the main unexpected risks in cave establishment, as the constructed void is big enough to disturb the equilibrium condition of the rock mass. Common geotechnical hazards include subsidence, mud rush, rock fall, caving stall, caving hazards, collapse, flying rock and other uncontrolled material movements. Air blasts must be encountered where the cave intersects any excavation.

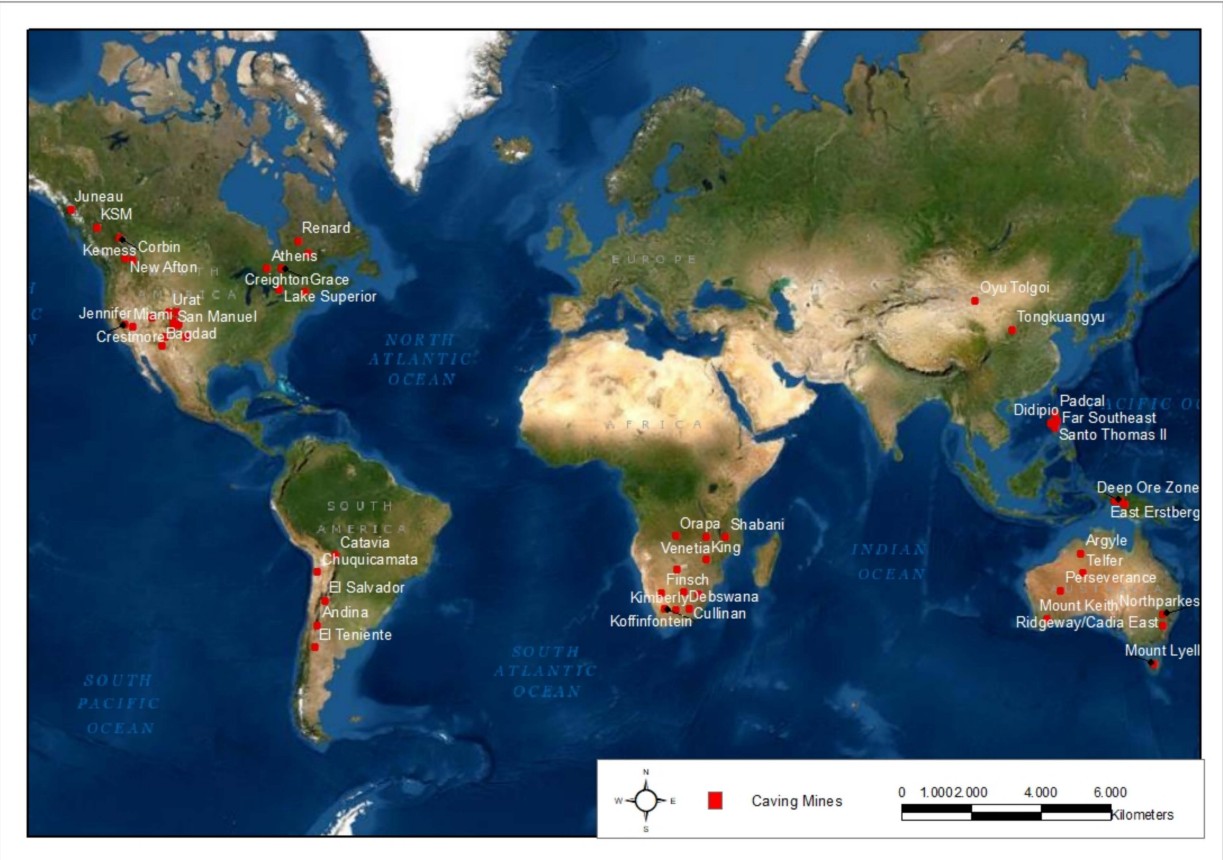

**Figure 3.** Distribution of caving mining globally compiled from previous studies, some of them listed in Table A1 (Appendix A).

It is also worth noting that the block caving applied on strong-hard-competent rock masses may face the problem of rock fragmentation. If the rock is deformed in coarse fragmentations, it might cause production loss and impede the continuity of ore flow at the draw point. Optimum fragmentation is a must to avoid material being stuck at draw points. One needs to understand the geological settings and structures of the in-situ rock: rock's strength, mineral composition, joint conditions, and natural fractures, to plan the mining strategies: pre-conditioning, undercutting, cave initiation, cave propagation, drawing and ore handling process, since these conditions achieve successful fragmentation and draws controlling. In some block caving, fragmentation problems could be avoided by a combination of comprehensive modeling of geological settings, statistical analysis and forecasting algorithms, digital imaging, laboratory testing, or numerical simulation for fragmentation modeling or dilution prediction [100–104].

Based on the last transformation and the practice of existing and developing block cave mining projects, the following sum up the new/updated typical orebody parameters requirement for block caving:

1. Ore body dimension

Suitable ore body types for this method are porphyry or pipe, large with thick, wide, tabular, or blocky dimensions. There are several applications on other types of deposits, i.e., sedex, VMS, stratiform, and others. However, the most common is porphyry.

2. Ore body and rock mass quality

As preconditioning had been a casual practice in many caving projects, there are no limited requirements for ore strength of the ore body. The upper limit of the uniaxial rock strength of competent rock mass to be mined using block caving is 300 MPa.

3.    Grade

Block caving is allowable for mining from low to high-grade ore bodies. The minimum cut-off grade varies depending on the specific site's engineering setting and other factors. However, the grade distribution of ore bodies should be relatively uniform. The exploited deposits have grade from 0.3% to >1.0% Cu.

4.    Depth

On the latest block caving mining projects, orebody lies at depths from 500 up to 2200 m below the surface.

Technical specifications for running block caving mines:

1.    Preparation method

Technical specifications of hydraulic fracturing and/or de-stress blasting are defined by ore body and rock mass quality. Undercutting blasting is continued until the cave propagation is achieved by a certain hydraulic radius depending on the in-situ rock strength.

2.    Undercutting strategy

The advanced undercutting trend is the most optimum strategy by compromising pre-undercutting and post-undercutting. This strategy allows only a limited excavation of the draw bell in the draw horizon before the undercutting process. The remaining development in extraction level is continued in the de-stress condition area.

3.    Mine Design

Both El Teniente and C-Cut are still used. There are no significant changes from the original version of the block caving mining method in undercut, production and draw bell configuration. Production rates accelerate to more than 80,000–130,000 tons per day.

4.    Mining equipment

Mucking equipment from loading in the draw point to haulage level consists of Load Haul Dumps. They have been operated autonomously in some super cave projects.

5.    Monitoring system

Micro-seismic monitoring, with its advancement in tomography technology, is a helpful and reliable tool for measuring damage and modelling stress distribution on the rock mass induced by block caving mines. It should provide a concise and precise description of the experimental results, their interpretation and the experimental conclusions that can be drawn.

## 4. Existing Caving Mine in Indonesia

### 4.1. Grasberg Caving Complex

Esrtberg prospective area was discovered by mining engineers, Jean Jacques Dozy and A. Colijn in 1936. However, this area was not explored in detail until Freeport discovered the report in 1960. An expedition led by Forbes Wilson and Del Flint rediscovered the Erstberg mineral deposit. By signing the Contract of Work (CoW) with the government of Indonesia in 1967, Freeport became the mining contractor for the Erstberg deposit. In 1973, Ertsberg was declared operational after completing the exploration and feasibility study. Grasberg deposits were discovered in 1988, three kilometers from the Erstberg mine [105]. Since the opening of Erstberg and Grasberg, several block caving mines have been operated by PT Freeport Indonesia (PTFI) to mine the ore body.

Orebodies on the CoW area of PTFI are extracted using surface and underground mining systems (Figure 4) [106]. Ertsberg East Skarn System (EESS) deposits were mined by block caving. Gunung Bijih Timur (GBT) block caving mine started production in 1980 through to 1993, from an elevation of 3474 m to 3626 m, recovering 60 million tons of ore. Intermediate Ore Zone (IOZ) or Erstberg Stockwork Zone block caving was in operation from 1994 until 2003, with total production of over 50 million ore. IOZ mine connected vertically with the GBT mine. The footprint of the IOZ orebody was 330 m long by 220 m

wide, with ore column heights of 150–220 m. DOZ mine block cave mine is located about 320 m below IOZ mine and about 1200 m below the surface.

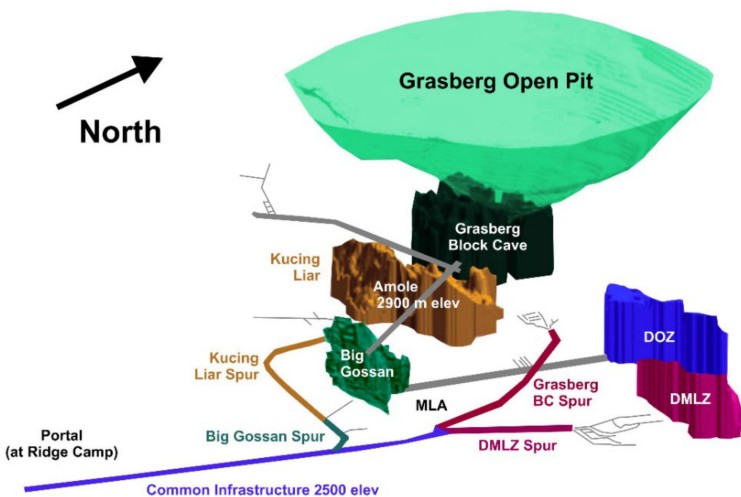

**Figure 4.** Ore bodies in Grasberg Block Cave Mine. There are five existing blocks in the Grasberg Block Cave Mine, i.e., Grasberg Block Cave, Kucing Liar, Big Gossan, DOZ, DMLZ. Each block is accessed through a spur towards the Portal at Ridge Camp.

DOZ block was at the elevation of 3126 m to 3476 m. The undercut level is 3146 m or 20 m above the production level. DOZ block cave was designed to be more than 1000 m long and 500 m wide. The mine had 857 draw points, designed in a column footprint of $15 \times 18$ m. There were three main specific geological units that were encountered in the DOZ mine: diorite and magnetite skarn, forsterite skarn and dolomite-marble and highly altered localized ore (HALO). They had a specific gravity of about 2.22–2.9, Rock Quality Designation (RQD) of 22–93%, Unconfined compressive strength (UCS) from 21–130 MPa, and Rock Mass Rating (RMR) 25–90, classifieds from poor to fair up to very good. The mining operation was completed after being targeted to produce 25 kilotons of ore per day [107].

Deep Mill Level Zone (DMLZ) is the deepest block caving below DOZ. DMLZ, located at elevation 3.125–2.590 msl, has a reserve of about 472 million tons, 0.85% copper and 0.72 g/ton gold. DMLZ targeted a diorite intrusion body bounded by dolomite. The diorite is similar that in Ertsberg, which was also altered by potassic phyllite and endo-skarn garnet intersected by quartz vein and anhydride [108–111]. Diorite in Extraction Level and Undercut Level has UCS 156.5 MPa and middle RQD 70–100% [59]. This strong, intact rock and competent rock mass characteristic make cave mining practice more challenging. DMLZ will have 700 active draw points in total production to complete the mine operation in 2041 [56].

Grasberg Block Cave (GBC) is the underground mine using the block caving method that continued production of Grasberg orebody after the Grasberg Open Pit Mine was completed in 2018 [109,110]. GBC began to develop in 2004 by drifting the first access. Caving was started by undercutting in September 2018 and draw-belling in December 2018. Grasberg orebody is a porphyry copper-gold deposit formed by a multiphase dioritic intrusion replenished in the center of a volcanic breccia complex. The mineralization is about 1600 m in length vertically and from about 200 m to over one kilometer in width. The extraction level using El Teniente-style layout with production panel drifts space of 30 m are at 2830 m elevation. The draw point spacing of 20 m was designed and resulted in a total of 2400 draw points. The undercut level was developed 20 m above the extraction level. The overall 700,000 m² large and about one kilometer in diameter footprint of the GBC orebody is sectioned into eight blocks. Operating multiple blocks simultaneously to meet the production target is exceptionally challenging due to its special condition, where a large

block cave sits below the largest open pit in an area of high rainfall. The mine closure of the cave was targeted in 2041 with a production of 130,000–160,000 tons per day. Character of rock mass in this cave mine is more complicatedly heterogenous than DOZ, classified as fair to very good ground and poor to fair ground regarding geotechnical domain. The first domain consists of 7 lithologies and has RQD 72–88% with UCS 80–140 MPa. The second comprises five (5) lithologies and has RQD 11–90% with UCS 5–80 MPa [56,105–113]. Kucing Liar block deposit, the next operated large block cave mine, was developed in 2022 and targeted to begin production in 2027.

No detailed technical notes related to blasting practice specifications at Grasberg Caving Complex were documented. Based on the underground ring blasting design (long hole blasting techniques applied in block caving), the most common diameter ranges from 64 to 115 mm. For draw belling, a fan of 102 mm diameter consists of 9 holes with the length of blast hole drilled in the roof range from 8 to 22.9 m. By burdening 2.5–2.6 m, it consumes about 58 to 188 kg emulsion (1.1 gr/cc) per hole: the smaller hole diameter, 89 mm, is used on a narrow inclined undercut ring to avoid dilution. The burden is designed to be closer, only ranging from 1.8 to 2.1 m, with 5 holes in a fan. It needs from 8.7 to 69 kg emulsion (1.0 gr/cc) per hole [114]. As the draw belling and undercutting blasting output, the broken ore was mucked from the draw point using Load Haul Dump (LHDs) operated remotely, then delivered via loading chute to a rail haulage system. Wire mesh and shotcrete were applied to support the draw point and undercut level. Blasting conducts for two rings (fan series) every day and results in about 80–100 kilotons of Cu-Au fragmented ore.

*4.2. Semi Caving in Pongkor*

The second case of a block cave mine in Indonesia presented in this paper is a small block caving modified from a typical underground mining method. Initially, all primary veins were mined using cut and fill as a common mining method. However, some ore mass collapses were found in sill drift during development. Therefore, in 2004, the semi-caving system was successfully applied in Pongkor underground gold mine. Given the classification of weak ore observed, caving mining was selected to be operated on those ore masses. Based on the in-situ rock strength, a small hydraulic radius of approximately 3.3 m was required for cave propagation. During application for four months with the undercut, the dimension caused by the continuous caving of the ore body was approximately 20 × 10 m and ore production increased to 95.7%. This case could be referred to by miners regarding how to modify the underground mining method as semi-caving if the ore body or rock mass is extremely weak for maintaining excavation [115].

**5. Mineralization Type and Potential Cave Mines in Indonesia**

In mine planning and design, ore body parameters are fixed technical and economic considerations. Mining method classified by ore body condition is the main aspect of planning and design. The mineralization style includes all the ore body's basic parameters. The geological setting and geotechnical characteristics of the ore body and the surrounding rocks are fundamental to mine design. This condition will influence tunnel support requirements and productivity. The depth below the surface, regional stresses, and geothermal gradients can all have significant impacts on aspects of mine design, performance, and cost. The commodity may also influence how the ore is mined, treated, or transported [65].

This means that the style of mineralization is a fundamental control in the life of a mine. This condition is in accordance with the laws and regulations in Indonesia. As written in the Indonesian Mining Law, 2020, mining is a part of or all of the activities stages to manage and undertake minerals or coal, which includes general study, exploration, feasibility study, construction, mining, processing and/or refinement, development and/or utilization, transportation and selling, and also post-mining activity.

This law had been detailed by the Ministerial Decree of Energy and Mineral Resources in 2018. It is explained within it that mine planning is arranged at the stage of the mine feasibility study. Mining method and system as an aspect of mine planning consists of

mining system (surface or underground), mining method, mine schedule, production rates, and life of the mine. The mining system must be suited to special and geotechnical conditions, ore bodies, mine environmental considerations, and mining technology. Based on this guidance, the inventory of ore deposits and their style of mineralization is the first step in assessing the appropriateness of block caving as a mining method in a specific site.

As the path of the world's ring of fire, the Indonesian archipelago was formed due to the interaction and collision of the gigantic crustal blocks of the Eurasian, Indian, Australian, and Pacific plates. The process was induced by ultrabasic rocks containing rich mineral resources distributed extensively in eastern Indonesia, while in western Indonesia most of the orebodies explored are associated with the active volcano-plutonic arc or the stable mass of the Sunda Shelf [116–119].

Indonesia's geological condition is promising since the magmatic arc is strongly associated with copper and gold mineralization. Gold mineralization in Indonesia was formed in the andesitic arc. The andesitic arc occurs in the Cretaceous range to the Pliocene (3–20 million years), especially in the Cenozoic age. At the time, Indonesia's plates started to experience subduction and actively generated a certain zonation of magmatic arcs. The identified gold deposits in Indonesia are copper-gold porphyry, skarn, high and low epithermal sulfidation system, sediment-hosted gold, gold-silver-barit and base metals deposits and kelian type, a transition from porphyry to an epithermal system [116–119]. Based on tectonic activity along the magmatic arc, the eastern part of Indonesia is dominated by porphyry and skarn formations, as well as a small percentage of hydrothermal sulfide deposits and hosted sediment. In Western regions of Indonesia, mineralization tends to be epithermal deposits. Low sulfidation was generated in relatively shallow areas of Sunda Land [119,120]. Regarding the relationship with magmatic arcs, deposits in Indonesia are related to andesitic magmatic arcs that formed rapidly during magmatic activity. This shows that this mineralization is related to the subduction of the ocean floor. Epithermal deposits were formed along the continental arc, which was the island arc joining the Sunda Shelf during the period of mineralization due to crust thickening and intensive elongation. Porphyry gold occurred both in the environment of the island arc and continental arc [119–121].

Porphyry and epithermal deposits form in the upper crust. They are related to sulfur and water-rich intermediate to silicic magmatic sources of hydrothermal fluids that move upward and produce extensive hydrolytic and alkali wall-rock alteration, quartz veins and sulfides. Figure 5 shows that the porphyry body is formed above magma chambers where fluids hydro fracture rock at 700–350 °C and pressures range from supra-lithostatic to supra-hydrostatic. The formation depth ranges from 2 to 10 km [119,120].

Block caving method had been implemented on these ore body genesis: kimberlite pipes, skarn deposits, porphyry deposits, asbestos deposits in peridotite rock, iron ore deposits and stratified sedimentary horizons. Thirty-one of 80 mines that used the block caving method were porphyry deposits, while 6 were skarn and porphyry-related deposits, including Grasberg Caving Complex in Indonesia [65]. It is proven that this type of porphyry deposit is very suitable for using this underground mining method, considering the relatively large shape of the ore body. Moreover, the block caving mines in porphyry deposits are practically a transition of the open pit above it, in terms of extending their mine life. Observing the existing copper and gold mines in Indonesia, the block caving mining method was only implemented economically in the porphyry type. In contrast, the other quartz veins were mined using the stopping method. Thus, it is highly recommended to explore porphyry deposits and declare them as prospective deposits for the application of the block caving method.

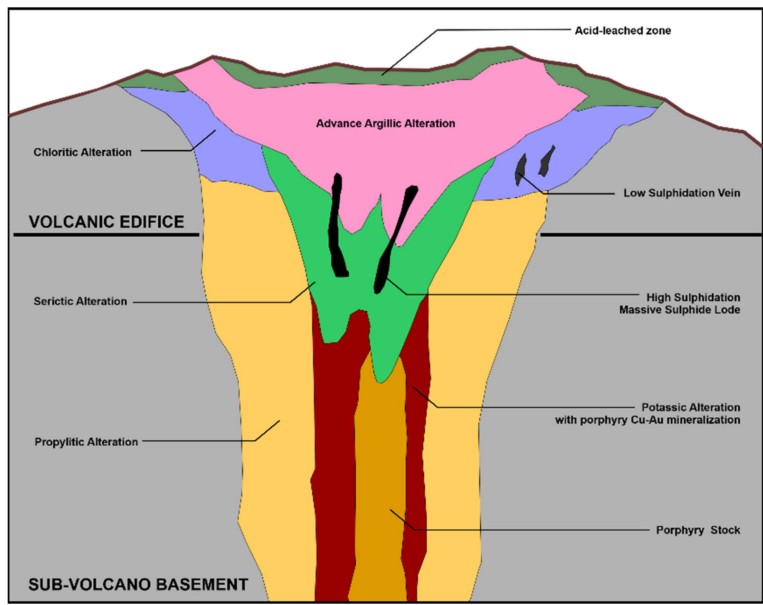

**Figure 5.** The schematic of porphyry ore body. Different colors indicate the different mineralization and alteration of the intrusion body.

Figure 6 shows the location of porphyry deposits in Indonesia. They are distributed mostly in eastern Indonesia. Tapada, Bulagidun, Tombulilato and Motomboto porphyry deposits were generated on Sulawesi East Mindanao Arc. Grasberg porphyry deposits and skarn system related were generated on Medial Irian Jaya Arc. The genesis of the Kaputusan porphyry deposit is associated with Halmahera Arc. Onto, Elang, Batu Hijau, Randu Kuning and Tumpang Pitu porphyry deposits are located in southern Indonesia, which is associated with the Sunda-banda Arc.

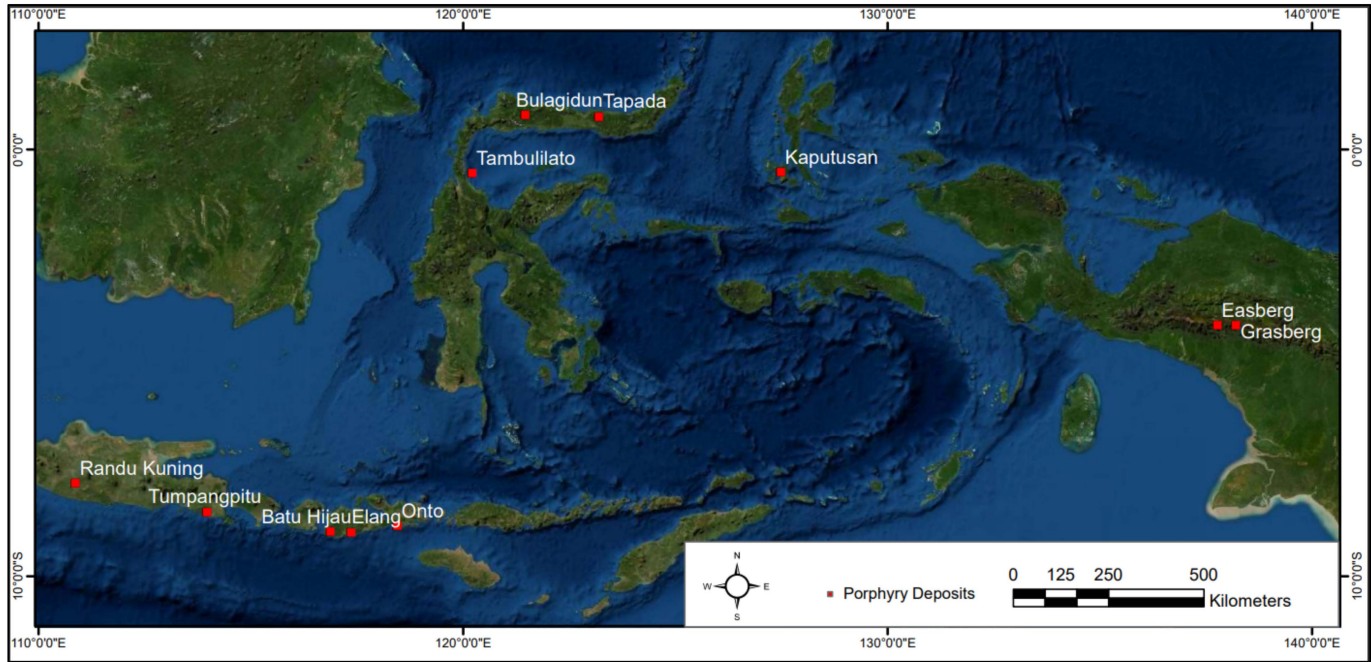

**Figure 6.** Porphyry deposits in Indonesia, compiled from previous studies in Table A2 (Appendix A).

Table A2 (Appendix A) shows that the style of mineralization of economic deposits in Indonesia is dominated by porphyry and quartz vein. Ore mineralization deposits in Indonesia are generally classified as primary and secondary deposits. A primary deposit

is formed from the magmatic process. Porphyry, stockwork, alteration zone, or skarn system are related to hydrothermal phases. In epithermal phases, gold enrichment fulfills the empty structure or voids in the rock mass, where Au/Ag/Cu/Mo quartz veins are concentrated. Secondary deposit such as alluvial placer transported and sedimented on river meander is extracted by local miners using artisanal or traditional mining methods.

A large dimension is a fixed requirement for implementing the block caving method. The large ore bodies explored in Indonesia with potential to be mined using the block caving method are the Onto deposit in Eastern Sumbawa, Cabang Kiri deposit in Gorontalo, Tumpangpitu-Tujuh Bukit Porphyry in East Java and Randu Kuning deposit. Most global block caving projects have been mining the Cu Porphyry orebodies. Indonesia needs to be concerned about performing a feasibility study on this ore body type.

Copper and gold mineralization and their associated minerals, such as silver and molybdenum, is related to magmatic arc. Generally, Indonesia's copper and gold mineralization results in various deposits of porphyry, high-sulfidation epithermal deposits, low-epithermal deposits, Au-Ag-Cu±base metal mineralization, skarn and sediment-hosted. Based on the tectonic events that occurred along magmatic arcs, eastern Indonesia is dominated by porphyry and skarn, some high sulfidation hydrothermal deposits and sediment-hosted. Western Indonesia has mineralization that consists of low-sulfidation epithermal deposits in the shallow Sunda arc [116–121].

Porphyry gold deposits can be formed on both island arc and continental arc [65,114–119]. Magmatic arcs in Indonesia where porphyry deposits are suspected to be found are Sunda-Banda Arc (Au-Cu porphyry), Aceh Arc (Cu-Mo porphyry), Central Kalimantan Arc (transition from epithermal to porphyry), Sulawesi-East Mindanao Arc (Au-Cu porphyry) and Central Irian Jaya Arc (porphyry and skarn orebody). Halmahera arc has not yet been explored. However, the mineralization type is hypothesized in the form of Cu-Au porphyry.

- Onto deposit

Onto deposit is a large Cu-Au deposit discovered in 2013 on eastern Sumbawa Island. Cu occurred as covellite and pyrite-covellite veinlets in a tabular block. The block dimension is at least 1.5 × 1 km and the vertical height is ≥1 km. In 2013, a diamond drill program tested an extensive advanced argillic alteration litho-cap within the Hu'u project on eastern Sumbawa Island, Indonesia. A very large and blind copper-gold deposit (Onto) was discovered, in which copper occurs largely as disseminated covellite with pyrite and as pyrite-covellite veinlets in a tabular block measuring at least 1.5 × 1 km, with a vertical thickness of ≥1 km. The porphyry intrusions were emplaced at shallow depth (≤1.3 km), with A-B–type quartz veinlet stockworks developed over a vertical interval of 300 to 400 m between ~100 and 500 m below sea level (BSL), 600 to 1000 m below the present surface, which is at 400 to 600 m above sea level. Although the greatest amount of copper occurs as para-genetically late covellite deposit during the formation of the advanced argillic alteration, approximately 60% of the resource at 0.3% Cu cut-off still occurs within the porphyry stocks, indicating that porphyry stocks are a fundamental control on mineralization [121].

- Porphyry deposits on Tambulilato

Porphyry Cu-Au mineralization on Tambulilato, North Sulawesi, is present at Cabang Kiri, Sungai Mak, Kayubulan Ridge and Cabang Kanan. Hypogene Cu-Au mineralization is typically associated with magnetite-bearing K-silicate assemblages, partially obliterated by sericite, illite and chlorite. Copper as chalcopyrite and bornite and Au show a positive correlation. Multiphase intrusions and alteration-mineralization events are commonplace. Biotite-bearing K-silicate alteration at Cabang Kiri and Kayubulan Ridge yields K-Ar ages of 2.93 ± 0.06 and 2.36 ± 0.05 Ma, respectively. Cabang Kiri possesses a gold-rich zone (>1.5 ppm Au) open at depth and the bulk of the mineralization at Sungai Mak is contained in a supergene chalcocite blanket [122].

- Tumpangpitu, Tujuh Bukit porphyry deposit

The inferred hypogene sulfide resource estimate for the Tumpangpitu Cu-Au-Mo porphyry deposit is 1.9 billion tons at 0.45% Cu, 0.45 g/t Au and 90 ppm Mo, equating to 28.1 Moz Au, 19 Blbs Cu and 400 Mlbs Mo at a 0.2% Cu cut off. The total measured, indicated, and inferred resource for the high-sulfidation Au-Ag oxide is 71.4 million tons at 0.80 g/t Au and 26.3 g/t Ag with 1.9 Moz Au and 60.3 Moz Ag at 0.3 g/t Au cut-off grade. The advanced exploration resulted in the Tujuh Bukit porphyry deposit having a total of 1.9 billion tons of inferred global resource of ore at an average grade of 0.45% copper and 0.45 g per ton of gold, containing 19 billion pounds of copper and 28 million ounces of gold. In early 2018, work began on the underground development of the exploration decline. It progressed encouragingly throughout the year toward developing a long-life and low-cost block cave porphyry copper and gold mine [123].

- Randu Kuning deposit

The Randu Kuning prospect is a part of the East Java Southern Mountain Zone, occupied mostly by plutonic and volcanic igneous rocks, volcaniclastic, siliciclastic and carbonate rocks. Magmatism-volcanism products were indicated by the abundance of igneous and volcaniclastic rocks of Mandalika and Semilir Formation and many dioritic intrusive rocks of the Late Eocene-Early Miocene magmatism. Porphyry Cu-Au and intermediate sulfidation epithermal Au-base metals mineralization at Randu Kuning have strong genetic correlation with the magmatism and volcanism processes. The mineralized dioritic intrusive rocks in the area are distributed at the center of the depression of an ancient volcanic crater. Many intermediate sulfidation epithermal prospect areas surround the Randu Kuning porphyry Cu-Au. Most mineralization, including porphyry and epithermal environments, is associated with quartz-sulfide veins. However, not all porphyry vein types contribute to copper and gold mineralization. The early quartz-magnetite veins (particularly A and M vein types) generally do not contain Cu-Au or are barren. In contrast, the later sulfide-bearing veins, such as quartz-sulfide (AB type) veins, chalcopyrite-pyrite (C type) veins and quartz-sulfides-carbonate (D type) veins are mineralized. Mineralization contains copper and gold deposits in the range of about 0.66–5.7 g/t Au and 0.04–1.24% Cu. On the epithermal level, mineralization is mostly related to pyrite+sphalerite+chalcopyrite+quartz+carbonate veins and hydrothermal breccias. The epithermal veins and breccia lead to the occurrence of silver, zinc and lead mineralization. It commonly contains around 0.4–1.53 g/t Au, 0.8–8.5 g/t Ag, 0.17–0.39% Cu, 0.003–0.37% Zn, 00089–0.14% Pb [124].

The block caving method is a promising underground mining method for mineral production in Indonesia, specifically for Cu-Au-Ag-Mo porphyry-type deposits. It is needed a comprehensive and high certainty of orebody knowledge. For thinner deposit shapes like quartz veins, semi-caving is reliable as an alternative method to modify the main method–usually stopping group type (cut and fill, shrinkage, sublevel stopping)–while the ore body has a weak or poor quality.

## 6. Opportunities and Challenges of Block Caving Method in Indonesia

The number of prospective ore bodies to be mined using block caving methods mentioned above will increase the exploration program. Presently, the new targets of exploration in Indonesia are determined by scholars and investors who spend their resources to study prospective areas on Sunda Arc, such as Tangse, Gunung Subang, Ojolali and Miwah [125–128].

### 6.1. Exploration

The exploration method should be by type of deposits to ensure the success of exploration activities. Valuable deposits in Indonesia are mostly porphyry and epithermal. The most useful geophysical techniques in exploration for these are high-resolution magnetics and electrical surveys. Airborne magnetic and electromagnetics surveys are fast and cost-effective, particularly in areas of rugged topography. Regional magnetics, gravity, remote sensed data, and topographic data can also identify major structures, intrusive

complexes and alterations. Radiometric surveys can be useful in mapping geology and alteration. Table A3 shows the recommended exploration methods classified by the type of deposits summarized [120]. Geophysical exploration methods benefit the prospecting and general exploration stage or scope study. After prospecting has been plotted, another detailed method should be conducted, such as detailed topography, core drilling and geophysical logging. Laboratory testing is conducted, including rock mass characterization and ore body quality. This aims to arrive at the best understanding of ore body conditions. Therefore, feasibility studies can be held using data with high confidence.

### 6.2. Cost

Investment capital cost to develop block caving method is expensive. For representation, the pre-feasibility cost of Tujuh Bukit Porphyry was about $58 million [123,129]. The development cost of GBC and DMLZ plus the infrastructure was $7.8 billion, taking place from 2008 to 2021 [130]. Ridgeway Deeps was developed at the cost of A$525 million [131]. Capital investment for Carrapateena block cave development of about A$1.2 billion to A$1.3 billion was weighted toward 2025 to 2027. It has been analyzed that incentive pricing of USD 3 to 3.5 per lb ore is required to upscale the currently proposed caving projects into the production stage [132]. Of course, this consideration is a special challenge for the investor. The joint venture between the government mining company and national or global world-class mining company investment hopefully will be helpful in accelerating the operation of the block caving method in Indonesia. Investment capital cost on block caving projects usually involves the construction of a block cave, underground crusher, automated remote loaders, modifications to the processing plant and deployment of the monitoring system. Raise caving, introduced recently, should be considered for its feasibility in reducing the cost of infrastructure development.

### 6.3. Operational

Block caving is a mining method with large productivity/mass mining. Remote-control systems for ore drawing and mucking are unavoidable to improve safety and pursue production targets. The deployment and mastery of this remote-control technology system should be supported by providing training/internships for prospective operators working in block caving mines. Mastery of the technology of controlling remote mining will be beneficial for the availability of Indonesian human resources competent in operating block caving mining systems. The possibility of adopting a block caving or even the raise caving method should be considered for optimum mining practice.

### 6.4. Environment

Block caving does not cause uncontrollable environmental impacts, such as changes in landscape on the surface, especially in block caving mines in strong deep rocks. In some instances where the deposited deposits result from an open pit mine, it is necessary to conduct a comprehensive study to establish the optimal crown pillar geometry [133,134]. In relatively shallow deposits, block caving can trigger subsidence at ground level. The maximum area, angle and subsidence depth should be predicted at the feasibility study stage and monitored during mining to ensure no facilities on the surface are affected [135,136]. Other risks to worry about during the operational phase of mining that may impact workers are rock stability, mud rush and air blast. With a good understanding of the geological structure, hydrogeology, and fluid mechanics within rock masses, the best mining designs and practices can be applied to control unexpected impacts. At the stage of detailed exploration, the three aspects must be appropriately characterized and the influence of mining on all three must be considered [137,138]. In addition to the development of science and technology, mastery of human resources to understand the condition of the rock mass is absolutely necessary in the context of implementing an environmentally friendly block caving mine in the future.

*6.5. Policy*

Indonesia has been experiencing a tortuous journey in mineral resource management. At the beginning of the independence era (1945–1965), President Soekarno spread a strong spirit of nationalism and anti-imperialism. In this era, the Indonesian government nationalized mining companies that the Dutch regime controlled then. Investment and aid from western countries were rejected [139]. In the New Order regime (1965–1997), foreign investments were facilitated by Mining Law 1967 with the mining work contract (Kontrak Karya) scheme. The reformation era had been inspired by the decentralization that had impacted the rapid growth of mining permits owned by local businesses. The government's lack of readiness for the monitoring system and the experience of mine owners in good mining practices left some "troublesome homework" until the present day. The biggest problem is that of the voids of former mines and their unfinished land arrangement, as their function is written in the mine closure documents. Finally, after almost half a century, Indonesia's government issued a new Mining Law in 2009 (Mineral and Coal Mining Law No.4 2009). It reorganized the superintendence of the mining industry, specifically, responsibility for carrying out product down streaming and prioritizing the fulfillment of domestic demand. After the legalization of the newest Mining Law (Mining Law No.3 2020–The Amendment of Mineral and Coal Mining Law No.4 2009) and Omnibus Law 2021, the Indonesian government remained open to global investors with commensurate authorization and benefits. This condition is more suitable for the investment climate and promises a brighter future for the mining industry in Indonesia. Therefore, there is a legal opportunity to apply the block caving method in Indonesia by experienced foreign companies.

*6.6. Social Geology*

Mining industries have specific characteristics: high return, high risk, high technology and high investment. Mining is an industry operating over long time periods, especially in the block caving method, where the development and production time is relatively long due to the large deposit volume. This requires engagement with community growth on the local scale and country development on a national scale. As echoed in the last decade, social geology has become a required aspect that must be fulfilled [139–141]. Besides being technically feasible and economically profitable, mining projects must also be socially acceptable. Every new investment that comes in must be able to make a real contribution to society. The issue has become a crucial and sensitive issue, especially due to the impact of the social gap issue since Grasberg and Erstberg reserves have been exploited until the present. The appropriateness of profit sharing between the mining company and Indonesia has been criticized. This special requirement is known as a "social license" to operate.

The new mode of block caving mining methods uses autonomously operated and machine-intensive means for producing a large volume of ore. It is possible to assume that this reduced need for human labor means that this mining project provides only a low positive impact on people's employment. In addition, Indonesian miners are not familiar with this method, so their participation will be restricted. The contribution of the block caving mining project must be realized in another form, for instance, technological transfer by training Indonesian workers to become advanced in the application of these methods. Indonesia should mirror settled mining countries such as Australia and USA, who have made block caving an underground mining method supplying a large total volume of their national ore production.

## 7. Conclusions and Recommendations

The new requirements of ore bodies, suitably mined using the transformed block caving method, have been successively identified. This method can be applied to a blocky ore body with a thickness of 200–800 m, various rock mass strengths until 300 MPa, from low to high (from 0.3% Cu until more than 1.0% Cu), but uniform in grade and at a depth from 500 to 2200 m. The technical specifications for running block caving mines,

including preparation methods, undercutting strategy, mine design, mining equipment and monitoring systems, have been synthesized. The potential ore bodies mined using the transformed block caving method have been recommended. The block caving method is promising for underground mining metal deposits in Indonesia, especially Cu-Au porphyry deposits. Although it is favorable for rock mass of various strengths and ore bodies at any grade level, a large dimension of the ore body is the fixed requirement. This condition is due to the need for fast and continuous production to comply with the specific pay-back period and settle the investment costs. Therefore, porphyry is the appropriate ore body type to be mined using this method. Considering the requirements and the successful practice of the block caving project in Grasberg Caving Complex as a role model, the Indonesian government has been suggested as the main responsible body, assisted by national or global investors. The suggestion is to concentrate on the detailed exploration of porphyry deposits and feasibility studies applying the method to the prospective ore bodies, i.e., Onto, Tambulilato, Tumpangpitu and Randu Kuning. In addition, exploration method, cost, operation, environment, mining policy and social geology are important aspects worth noting. The infrastructure development stage of block caving needs 3–5 years until continuous production. The existing block caving projects have a life for production ranging from 12 to 35 years. Thus, a long-term concession is proposed, of about 15 until 40 years, covering development and operational production stages.

**Author Contributions:** Conceptualization, S.M., R.K.W. and D.P.S.; methodology, S.M., R.K.W. and D.P.S. and S.; software, S.M., W.H. and E.R.; validation, R.K.W., D.P.S., S., G.M.S. and E.R.; formal analysis, R.K.W., D.P.S., S.M., G.M.S. and S.; investigation, S.M., R.K.W., D.P.S., W.H.; resources, R.K.W., D.P.S., S.M. and G.M.S.; data curation, S.M., E.R., D.P.S. and G.M.S.; writing—original draft preparation, S.M., R.K.W., D.P.S.; writing—review and editing, S.M., R.K.W., D.P.S., S., R.R.S.F.; visualization, S.M., W.H.; supervision, R.K.W. and D.P.S.; project administration, S.M.; funding acquisition, R.K.W., D.P.S., S.M. All authors have read and agreed to the published version of the manuscript.

**Funding:** This study was supported by the Indonesian Education Scholarship from Puslapdik and LPDP, the Ministry of Education and Culture, awarded to Sari Melati.

**Data Availability Statement:** The main text includes all the data supporting this article.

**Acknowledgments:** We appreciate all miners, scholars, and practitioners in the block caving method for their devotion to developing this mass underground mining method. We would also like to express our gratitude to the researchers in our groups: Rock Mechanics, Global Geophysics and Earth Resources Exploration. Their invaluable knowledge and competence assisted us in dealing with this new issue. We thank the staff and technicians at Geomechanics and Mining Equipment Laboratory, Vulcanology and Geothermal Laboratory, Geophysical Instrumentation and Electronics Laboratory, Mineralogy, Microscopy and Geochemistry Laboratory, for their assistance in providing us with the resources needed to run the research. This study is supported by research funding from ITB entitled *Quantification of competent rock mass on block caving mining using micro-seismic monitoring and numerical modeling*, awarded to Ridho K. Wattimena.

**Conflicts of Interest:** The authors declare that the research runs without any commercial or financial relationships that could be construed as a potential conflict of interest.

## Appendix A

**Table A1.** Block caving projects in the world.

| Mines | Ore Bodies and Types of Mineralization | Rock Mass | Depth | Production Rates and Reserves | Footprint | Mine Design | Equipment | Time | Ref. |
|---|---|---|---|---|---|---|---|---|---|
| Argyle, Australia | Diamond pipe. Volcanic vent intrusion of magmatic lamproite and lamproitic tuff. | Granite, dolerite, basalt and metamorphosed quartzite and mudstone. UCS 35–104 MPa, RMR 45–59 | $\sigma_1 = 2\sigma_v$, $\sigma_2 = 1.5\sigma_v$, $\sigma_3 = 0.027z$ | 18,000 tpd (Lift 1). It has produces 800M carats | 75,000 m² | an advanced undercut technique using a W-incline undercut design | Real-time LHD dispatch | 2008 (undercutting), 2015 (development complete), 2020 (final production) | [142,143] |
| Cadia East, Australia | Monzonite porphyry, Au-Cu porphyry deposits | Andesit, monzonite, quartz. UCS 132–140 MPa, E 65–67 GPa, FF < 15. | 63:42:36 @1200; 72:48:41 @1400; | 26 Mta | Width orebody 700 m | El Teniente, Drawbell Spacing: 32 × 20 m | Load-Haul-Dump (LHD) operation | 2000 (production) | [47,77,99] |
| Carrapateena, South Australia | copper-gold deposit | brecciated granite complex | 500 m | 10,000–120,000 t of copper and 110,000–120,000 oz | 70,000 m² | El Teniente, draw point spacing of 32 m × 22 | LHD., Jaw Gyratory crusher, crushed-ore-bin, conveyor system | 2020 (Prefeasibility study), 2026 (Production), 2045 (final year) | [132,144–146] |
| Cullinan, South Africa | Kimberlite pipe | UCS Kimberlite: 80–130 (Grey), 73–193 (Hypabyssal) Mpa; UCS Country rock: 140–220 (Norite), 60–240 (Metasediments), Hydraulic radius: 30; Mining Rock Mass Rating: 30–50 (grey), 25–35 (contacts, internal dykes and shear zones), 40–60 (Hypabissal) | 630–732 mbs | 3.9 Mt/a; Reserves: 38.6 Mt, grade 38.8, 14.97 Mt | 32 ha | Centenary-Cut; Undercut tunnels 4 m wide and 4 m height, 16 m spacing; Extraction level 4.2 m wide by 4.2 m high, spacing 16–18 m; Tunnel spacing in the production level of 32 m; Drawpoint spacing 18 m | Tamrock Toro–LHD | 1980–2037 (operation) | [147] |

**Table A1.** *Cont.*

| Mines | Ore Bodies and Types of Mineralization | Rock Mass | Depth | Production Rates and Reserves | Footprint | Mine Design | Equipment | Time | Ref. |
|---|---|---|---|---|---|---|---|---|---|
| El Teniente, Chile | copper-molybdenum deposit | Andesite, dacite, diorite, braden pipe; UCS 120, 110, 140, 90 MPa; RMR 53–59, 59–66, 64–66 | 2200 m. $\sigma_1 = 0.0328z + 16$, $\sigma_2 = 0.0283z + 5$, $\sigma_1 = 0.0265z$ | Productions 140,000 tons per day. Measured resources 1128 million tons 0.985%Cu | 500–800 m | El Teniente, hydraulic radius 26 m | Load-Haul-Dump (LHD) | 1997 (pre-undercutting), 2032 (planned final production) | [66,148] |
| Grasberg, Indonesia | Cu-Au Porphyry, Skarn | Fair to the very good ground: 80–140 MPa; Poor to fair ground: 5–80 MPa | 1200 m | 60,000–100,000 tpd; 160,000 (planned for 2026) | Area: 700,000 m² | El Teniente, Drawbell Spacing: 20 × 30 m | Load-Haul-Dump (LHD) operation, rail haulage system | 2004 (construction), 2018 (production) | [59,109,110, 112,149,150] |
| Jwaneng, Botswana | diamond-bearing kimberlite complex | Sand, calcrete, laminated shale, carbonaceous shale, quarzitic shale, chert pebble conglomerate-bevets, carbonaceous shale and dolomite. UCS 25 MPa (weak kimberlite), >250 MPa (very competent dolomite) | ~1000 m; $\sigma_1 = 0.9$–$1.1\sigma_v$, $\sigma_2 = 0.5\sigma_v$, $\sigma_3 = 0.027z$ | No data found | No data found | No data found | No data found | 2032 (construction) | [151,152] |
| Northparkes Mine, Australia | Trachyandesites (Volcanics) and finger-like monzonite porphyry (MP) intrusions, potassic alteration and occurs predominately in stockwork quartz veins. | Gypsum and quartz. MRMR's in Lift 1 ranged from 33 to 54. | >800 m | 16,000 tpd (E26 Lift 1, Lift 2, Lift 2 N); 18,000 tpd (E48 Lift 1). Reserves 27 million tons of Ore. | Width vein 200 m, height 800 m. 196 meters long by 180 meters wide. | Northparkes layout style, Hydraulic radius 20–25 | Load Haul Dump | 2002 (production) | [48] |

**Table A1.** *Cont.*

| Mines | Ore Bodies and Types of Mineralization | Rock Mass | Depth | Production Rates and Reserves | Footprint | Mine Design | Equipment | Time | Ref. |
|---|---|---|---|---|---|---|---|---|---|
| Oyu Tolgoi, Mongolia | copper-gold-molybdenum mineralization | volcanic and quartz monzo-diorite (QMD); Dacite tuff, breccia (IGN), basalt flows and minor volcaniclastic strata (Va). Dikes: rhyolitic, hornblende biotite andesite, dacite and basalt. MRMR < 20 | 1385 m | 95,000 tpd (Hugo North Lift 1) | Hugo deposit height 900 m, length 1.8 km, width 500 m. | El Teniente draw point layout on 31 × 18 m spacing | underground trucking system, gyratory crushers, conveyor system, concentrator | 2015 (construction), 2020 (production) | [57,153] |
| Padcal, Philippines | Cu-Au Porphyry | 0.18% Cu, 0.27 g/t Au; 56 Mlbs Cu, 166.700 oz Au | No data found | Production 70,000 m$^2$ | No data found | No data found | No data found | 2020 (exploration) | [154] |
| Palabora, South Africa | Magmatic-hydrothermal deposit. | Carbonatite, 139 MPa (intact), 111 MPa (rock mass) | 1200–1800 m. | Production 30,000–82,000 tpd; Reserve 960 Mt | 250 × 650 m | off-set herringbone style, 20 cross-cut | LHD, crusher | 2000–2014 | [50,51,104] |
| Ridgeway Deeps, Australia | Au-Cu porphyry | Cadia Valley Monzonite (93–155 MPa), Forrest Reef Volcaniclastics (87–150 MPa) and Weemalla Sediments (88–144 MPa). Average density 2.85 t/m$^3$ | 1100 m. $\sigma_1$ = 65 MPa, $\sigma_2$ = 47 MPa, $\sigma_3$ = 32 MPa | 101 mt at 1.8g/t Au and, 0.38% Cu for 2.6 Moz Au and 380 kt Cu | 500 × 200 m$^2$ | Offset Herringbone layout and consists of 15 extraction drives, 250 drawpoints | Load haul Dump | 2005–2017 (production) | [50] |
| Shabanie, Zimbabwe | Asbestos | Dunite Sill intruding Precambrian Gneisses | No data found | No data found | No data found | No data found | No data found | 1970 (production) | [146] |

**Table A1.** *Cont.*

| Mines | Ore Bodies and Types of Mineralization | Rock Mass | Depth | Production Rates and Reserves | Footprint | Mine Design | Equipment | Time | Ref. |
|---|---|---|---|---|---|---|---|---|---|
| Stornoway Diamonds' Renard Mine, Quebec, Canada | Kimberly pipe | Pyroclastic, granitoid and gneissic host rock, UCS 4.5–26 MPa | 600 m. $\sigma_1 = 0.9$–$1.1\sigma_v$, $\sigma_2 = 0.5\sigma_v$, $\sigma_3 = \rho.g.h$ | 3000–5000 tpd | 225 m | Herringbone. Drawpoints are 5.3 m wide, distance between center 15 m | Load haul Dump | 2018 (production) | [55] |
| Lvivvuhillia SE Mine, and Ukraine | Coal, carbonous formation | Sandstone, Argillite, Aleurite | Sandy shale 23.2–31.1 MPa | 100 ktons per months | its average mining thickness is 1.24 m. | 10.3–10.6 m$^2$ for boundary entry | Coal shearers, Scraper, Oil-pumping station | 2020 (production) | [25,26] |
| the 10th Anniversary of Kazakhstan's Independence Mine, and Kazakh-stan | Chromite deposits | Peridotite and Serpentinite, UCS 17.1–64.5 MPa | Depth 900 m, $\sigma_1 = \sigma_3 =$ and $\sigma_z = 24.8$ MPa | No data found | 180 m | Undercut-caving system, Drawpoint spacing 12–24 m | No data found | Development (2021) | [155] |

**Table A2.** Ore deposit and type of mineralization in Indonesia.

| Location | Ore Genesis | Type | Size, Dip | Grade, Volume | Rock Mass | Mining Method | Status | References |
|---|---|---|---|---|---|---|---|---|
| Awak Mas, Latimojong, South Sulawesi | Hydrothermal | albite-ankerite-pyrite alteration halo | up to ~75 m width | Indicate and inferred resource of 38.4 Mt at 1.41 gr/t Au~1.74 Moz Au | Phyllite and schist. | - | Exploration | [156] |
| Batu Hijau | Epithermal | Porphyry | a zone 300 m × 900 m containing > 0.3 wt % Cu | > 0.1 wt % Cu, > 0.1 wt % Cu, Mo (> 30 ppm) | Diorite, metavolcanic rock | Open Pit, Block Caving (in panning) | Production Planning | [157,158] |
| Beruang Kanan, Kalimantan Tengah | Epithermal | quartz vein, porphyry | vein direction is N 312° E/43° | Not explore yet | Dasite, diorite, silica sand | - | 2017 (Exploration) | [159] |
| Bombana, Southeast Sulawesi | Secondary (placer) in Langkolawa in Wumbubangka derived from orogenic gold | Gold-bearing quartz vein | 2 cm–2 m | grades <0.005 g/t to 134 g/t | mica schist, phyllite, metasandstone and marble) | Placer mining -artisanal and small-scale gold mining | 2011 (study) | [160,161] |
| Bulagidun | Hydrothermal | a copper, gold and tourmaline bearing porphyry and breccia system | up to 500 m lateral distance, veins up to 2 m true width, | more than 14.4 Mt at 0.68 ppm Au and 0.61 wt.% Cu | early diorite to quartz diorite to late tonalite and post-mineral andesitic dykes. | - | Geological Study | [162] |
| Cibaliung, Banten | Epithermal | Quartz vein | Dyke 1 to 120 m wide, 20 to >300 m long. | 1.3 Mt 10.42 g/t Au, 60.7 g/t Ag 3 g/t cut-off; 435,000 ounces of Au and 2.54 Mounces Ag | UCS 16.85 MPa, Tensile strength 0.69 MPa | Cut and fill | 2001 (Exploration); 2010 (production | [163,164] |
| Cikidang (Cikotok) | Low sulfidation epithermal adularia | Quartz vein | Thickness 0.7–3 m, dip 60–86° | 74.9 g/t Au, 1.2–225 g/t Ag | Lapilli tuff, breccia andesite, claystone, limestone, Sandstone | Underhand stall-stopping method | 1998 (production), | [165] |
| Elang | Epithermal | Porphyry | undescribed | 300 t Au, >5 Mt Cu | Volcanoclastic and esitic | - | Exploration | [123,129] |

**Table A2.** *Cont.*

| Location | Ore Genesis | Type | Size, Dip | Grade, Volume | Rock Mass | Mining Method | Status | References |
|---|---|---|---|---|---|---|---|---|
| Ertsberg | Contact metasomatism | Skarn system | length > 1.1 km, 4–60 m thick, depth >700 m | 2.69 percent Cu, 1.02 g/t Au and 16 g/t. | dolomitic sediments | Block Caving | Production | [105] |
| Gosowong, Halmahera | Epithermal | Quartz vein, porphyry | Thickness 30–40 m, dip 35–70° | 0.99 million metric tons (Mt) at 27 g/t Au and 38 g/t Ag | Volcaniclastic and pyroclastic | Open pit | 1996 (exploration) | [118] |
| Grasberg | Contact metasomatism | Porphyry | 1.2 km (pit) | over 32 Mt of Cu and 3 kt of Au | Diorite, limestone | Open Pit, Block Caving | Production | [56,109,110, 150] |
| Gunung Subang, West Java | Epithermal | Gold-bearing minerals | 0.01–0.2 m; 40–81° | Au 0.22–14.49 ppm, Ag 17–21.40 ppm, Cu 8.25–34515 ppm, Pb 107.69–2226 ppm, 35.36–7335 ppm | Andesite, tuff, breccia | - | 2018 (Prospection) | [126] |
| Kelian, East Kalimantan | Epithermal | Au-Ag mineralization | 0.25–5 m | 240 t Au | Rhyolite | Open pit | 2003 (Mine closure) | [166] |
| Kencana | Epithermal | Au deposit | Thickness 12 m, 45° | 39 g/ton gold | RMR 25–55 and esite lavas, | Underhand cut and fill | Production | [167] |
| Malala, Northwest Sulawesi | Hydrothermal | fluorine-poor (quartz monzonite or differentiated monzogranite) class of molybdenum deposits | 50 m | estimated resource of 100 Mt at 0.14% MoS. | granites and granodiorites | - | 1993 (Geological study) | [168] |
| Miwah, Aceh | Hydrothermal | high-sulfidation Au–Ag deposit | >60° | inferred total resource of 3.13 million oz (Moz) of Au at a cut-off grade of 0.2 g/t Au | silicified rocks, breccia | - | 2019 (Prospection) | [128] |

**Table A2.** *Cont.*

| Location | Ore Genesis | Type | Size, Dip | Grade, Volume | Rock Mass | Mining Method | Status | References |
|---|---|---|---|---|---|---|---|---|
| Ojolali, Lampung | Epithermal | Tambang Vein:Ag-Au intermediate sulfidation deposit; Bukit Jambi Vein: low sulfidation Au-Ag deposit | <50 m, ~50° | inferred resource 167 g/t Ag and 0.7 g/t Au, forms a total of 40 Moz Ag and 170,000 oz Au | Basalt and esite | - | 2014 (Geological study) | [127] |
| Pani JV Project, Hulawa, Gorontalo | Hydrothermal | Open vein and breccia | No description | Resources 72.7 mt, 0.98 g/t, 2.3 mlb Au | UCS 21.42 MPa, UTS 2.06 MPa | Breccia, granodiorite and dasite | Conceptual study (2020) | [123,129] |
| Pongkor | Hydrothermal alteration | Vein | Thickness 2–24 m | 2.1 million metric tons at 13.63 ppm gold and 163.24 ppm silver (proven ore reserve) | volcanic breccia, lapilli tuff and esite lava and siltstone | Cut and Fill Stopping, Semi Caving | Production | [115–117] |
| Tambulilato: Cabang Kiri, Sungai Mak, Kayubulan and Cabang Kanan | Hydrothermal | Poprhyry (Cabang Kiri, Sungai Mak, Kayu Bulan, Cabang Kanan), high-sulfidation epithermal Au-Ag (Motomboto); low-sulfidation epithermal Au-Ag (Kaidundu) | Various wide of veins and porphyry | 392.3 million tons, 0.49%Cu, 0.43 g/t/Au, 1.65 g/t Ag. | Dacite, vulcanic, diorite | Stopping underground mining | Production (until 2052) | [122] |
| Tangse, North Sumatra | Hydrothermal | Cu-Mo porphyry deposit | Not explored yet | Not explore yet | Diorite | - | 2018 (Prospecting) | [125] |
| Toguraci, Halmahera | Epithermal | Quartz vein, porphyry | - | 0.41 Mt, 27 g/t Au | Andesitic lava, UCS 80 MPa | Under Hand Cut and Fill (UHCF) and Open Stope (Sub Level–Blind Stope) | 1996 (exploration) | [169] |

**Table A2.** *Cont.*

| Location | Ore Genesis | Type | Size, Dip | Grade, Volume | Rock Mass | Mining Method | Status | References |
|---|---|---|---|---|---|---|---|---|
| Tujuh Bukit | Hydrothermal | Porphyry | described | Inferred resources 1.9 bt, 0.45% Cu, 0.45 g/t, 8.7 mt Cu, 28 mlb Au | sedimentary and andesitic volcanic rocks | Open Pit | Production (2021) | [123,129,170] |
| Tumpangpitu, East Java | Epithermal | Porphyry | Mineralization > 800 m | 1.9 Gt, 0.45% Cu, 0.45 g/t Au | Diorite and esite, breccia | - | Exploration | [123,129,170] |
| Underground Tujuh Bukit | Hydrothermal | Porphyry (high level porphyry copper-gold-molybdenum deposit (sulfide) | | Inferred resources 1.9 bt, 0.45% Cu, 0.45 g/t, 8.7 mt Cu, 28 mlb Au | sedimentary and andesitic volcanic rocks | Underground mining (undetermined) | Pre-feasibility study (2021) | [123] |
| Wetar, Pulau Wetar, Southwestern moluccas | Volcanic- hosted massive sulfide (VMS) | Primarily pyrite | ~150,100.70 m and ~1,209,030 m, | 20 mt, 38% S, 33% Fe, host Cu, Au, Ag, Zn | Basaltic and andesite | Open pit | Production (2010) | [171] |

**Table A3.** Recommended exploration methods for the ore body type.

| Type of Deposit | Characteristic | Recommended Exploration Methods & Tools | Rationalization |
|---|---|---|---|
| Porphyry Cu-Au deposits | Commonly associated with magnetite that can produce strong discrete magnetic anomalies. Strong charge abilities due to sulfides are typically associated with porphyry systems. | High-resolution magnetic survey | Porphyry is usually within a zone of magnetite-destructive alteration. Magnetic surveys are also valuable for defining regional structure and geology in the porphyry environment. |
| | | Gravity, radio metrics, remote sensing and topography | Mineralization and clay-pyrite alteration can produce strong anomalies and late-stage and post-mineral intrusions can be mapped as low chargeability within the system. These systems may be more conductive than the host rock because of clay-pyrite alteration and sulfide veining and airborne electromagnetic can be helpful in locating and defining their extent. |
| High sulfidation epithermal system | Gold is commonly associated with massive silica alteration. | Resistivity and airborne electromagnetic survey | This alteration results in resistivities in the order of thousands of ohmmeters compared with background resistivities of tens of ohm-meters in argillic and propylitic alteration. Alteration in high sulfidation epithermal deposits is magnetite destructive over a large area, although it does not appear to have a large vertical extent as the subdued characterization of the underlying lithologies can be observed. |
| Low sulfidation epi-thermal system | Gold in this deposit is in thin quartz veins associated with major structures. Some deposits are associated with broad zones of magnetite destruction, which is apparent in the regional magnetics. | High-resolution magnetics, resistivity surveying | The alteration associated with the veins is magnetite destructive and high-resolution magnetics can be beneficial and cost-effective technique to map the structures and alteration. Generally, the high resistivity zones are due to silicification are coincident with the structure identified in the magnetics. |

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
