# Peer review of "Block Caving Mining Method: Transformation and Its Potency in Indonesia"

_energies, doi:10.3390/en16010009_

Round 1
Reviewer 1 Report
The manuscript is very interesting and contains valuable practical information on potential mining technology that can be applied under geological conditions in Indonesia. Many detailed data on metal ore mineralisation, known mining methods, and international experience in these technologies, emphasising the block caving and its modifications as the most appropriate one for the specific case, economic considerations on profitability of the future investment have been collected based on wide literature overview and supported by general discussion. The content of the manuscript is similar to that of a feasibility study. The knowledge contained here may be useful for engineers, students, and scientists, searching for any knowledge related to mining engineering, which is the most important value of the manuscript. It is also worth underlined that the idea of the article is to find the most effective, save and modern technology that is necessary to secure raw materials for high-tech development, including energy sufficiency in the world. This is not typical research work.
Author Response
We thank the reviewer for the appreciative comment that the manuscript's purpose/message is well understood. We updated this latest manuscript per the reviewers' recommendations by adding some technical explanations and more detailed conclusions. In sub-section 3.3. Requirements and Technical Specification Block Cave Mines in the World, we add the indicative values of the thickness of the ore body, the depth from the surface, and the grade of ore. We also add more information related to blasting and supporting practice applied typically on block caving in sub-section 4.1. Grasberg Caving Complex. Hopefully, these quantitative specifications will provide a better understanding of this method. Ultimately, we hope this comprehensive study of the block caving method can be useful for all mining engineers, scientists, and academics.

Reviewer 2 Report
Reviewer Comments
Paper title: Block Caving Mining Method: Transformation and Its Potency in Indonesia
The present manuscript summarized the growing technology in managing the block caving method based on existing block cave mines projects in all countries. The discussions of the transformation and potency of block caving in Indonesia are based on four aspects, i.e., technological advancement in block caving, geologic and tectonics analysis of Indonesia, existing operations, and non-technical aspects.
A manuscript has a practical application and also provides important theoretical for the next studies.
The paper can be accepted for publication after providing the corrections mentioned below.
Point 1. The abstract section sounds unclear. The abstract should follow the MDPI style of structured abstracts: Background (place the question addressed in a broad context and highlight the purpose of the study); Methods (describe briefly the main methods); Results (summarize the article's main findings); Conclusion (indicate the main conclusions or interpretations).
Point 2. Keywords need to be modified. Please use words not combinations of words or phrases.
Point 3. What does the keyword “mining country” mean?
Point 4. It is the review paper. You should indicate it on the left corner of the first page.
Point 5. In the Introduction section, an enhanced literature review is required. For this study, the authors have used only 15 literature sources. It seems insufficient for such type of research. You should provide more information about the Caving Mining Method around the world.
Point 6. You are kindly asked to indicate in your paper that caving method is also used in Ukraine Chile and Kazakhstan as described in references below
Shavarskyi, Ia., Falshtynskyi, V., Dychkovskyi, R., Akimov, O., Sala, D., & Buketov, V. (2022). Management of the longwall face advance on the stress-strain state of rock mass. Mining of Mineral Deposits, 16(3), 78-85.
Orellana, L. F., Castro, R., Hekmat, A., & Arancibia, E. (2017). Productivity of a continuous mining system for block caving mines. Rock Mechanics and Rock Engineering, 50(3), 657-663.
Matayev, A., Kainazarova, A., Arystan, I., Abeuov, Y., Kainazarov, A., Baizbayev, M., Demin, V., & Sultanov, M. (2021). Research into rock mass geomechanical situation in the zone of stope operations influence at the 10th Anniversary of Kazakhstan’s Independence mine. Mining of Mineral Deposits, 15(1), 103-111.
Point 7 It will be great if the authors show some description in context – Why it is important to conduct this study?
Point 8. The novelty of the paper must be highlighted in the conclusions section.
Point 9. Please consider the suggested research in your paper when compared underground mining methods:
1). Ibishi, G., Yavuz, M., & Genis, M. (2020). Underground mining method assessment using decision-making techniques in a fuzzy environment: case study, Trepça mine, Kosovo. Mining of Mineral Deposits, 14(3), 134-140. https://doi.org/10.33271/mining14.03.134
Important issue: According to UBC approach six alternatives – Block Caving, Cut-and-Fill Stoping, Sub-level Caving, Sub-level Stoping, Square Set Stoping, and Top Slicing have been considered as technically feasible alternatives. Results shows that cutand-fill stoping method is the optimal mining method for deep excavation mining levels.
2). Wu, H., Yin, Z., Zhang, Y., Qi, C., Liu, X., & Wang, J. (2022). Optimization of underground coal mining methods based on life cycle assessment. Frontiers in Earth Science, 933. https://doi.org/10.3389/feart.2022.879082
Important issue: A summary comparison of the single scores of different mining methods suggests that the environmental burden of pillarless mining is the lowest, and as the width of the coal pillar gradually increases, its single score shows a trend of increasing and then decreasing. Therefore, the single score of non-pillar mining is the lowest compared to that of other mining methods and can be the optimal mining method.
Point 10. In general, the presented article leaves a positive impression and, after eliminating these comments and taking into account the recommendations made, it can be recommended for publication in the journal "Energies".
Author Response
Reviewer 2:
Reviewer
The present manuscript summarized the growing technology in managing the block caving method based on existing block cave mines projects in all countries. The discussions of the transformation and potency of block caving in Indonesia are based on four aspects, i.e., technological advancement in block caving, geologic and tectonics analysis of Indonesia, existing operations, and non-technical aspects. A manuscript has a practical application and also provides important theoretical for the next studies.
The paper can be accepted for publication after providing the corrections mentioned below.
Point 1. The abstract section sounds unclear. The abstract should follow the MDPI style of structured abstracts: Background (place the question addressed in a broad context and highlight the purpose of the study); Methods (briefly describe the main methods); Results (summarize the article's main findings); Conclusion (indicate the main conclusions or interpretations).
Response to the reviewer
We thank the reviewer for the appreciative comment. The reviewer's purpose/message for this manuscript has been well understood and received. We have updated this latest manuscript as per the reviewers' recommendations by adding some technical explanations and more precise conclusions. We hope this comprehensive study of the block caving method can be helpful for all mining engineers, scientists, and academics.
Below is the revised abstract following the MDPI style of structured abstracts: background, methods, results, conclusions.
The block caving mining method has become increasingly popular in the last two decades. Meanwhile, Indonesia has several potential ore bodies which not yet determined suitable mining methods. We reviewed references to block caving mining projects worldwide and the potency of metal deposits in Indonesia. We determine the new requirements of ore bodies suitable mined using the transformed block caving method. This method can be applied on the blocky ore body with a thickness of 200-800 meters, various rock mass strengths until 300 MPa, from low to high (from 0.3% Cu until more than 1.0% Cu) but uniform grade, and a depth from 500 to 2200 meters. We also synthesize technical specifications for running block caving mines, including preparation methods, undercutting strategy, mine design, mining equipment, and monitoring systems. Considering the requirements and the successive practice of block caving project in Grasberg Caving Complex as a role model, we suggest the Indonesian government should concentrate on the detailed exploration of porphyry deposits and the feasibility studies of applying the method on the prospective ore bodies i.e., Onto, Tambulilato, Tumpangpitu, and Randu Kuning. In addition, exploration method, cost, operation, mining policy, and social geology are important aspects worth noting.
Reviewer
Point 2. Keywords need to be modified. Please use words not combinations of words or phrases.
Response to the reviewer
We use the new keywords as follows:
Caving; Indonesia; mining; porphyry; underground
Reviewer
Point 3. What does the keyword “mining country” mean?
Response to the reviewer
A mining country refers to a country that relies on the mining sector as one of its country's foreign exchange sources.
In the revised version of our manuscript, the “mining country” has been excluded from the keywords and replaced by porphyry.
Reviewer
Point 4. It is the review paper. You should indicate it on the left corner of the first page.
Response to the reviewer
We had written “Review” on the left corner of the first page.
Reviewer
Point 5. In the Introduction section, an enhanced literature review is required. For this study, the authors have used only 15 literature sources. It seems insufficient for such type of research. You should provide more information about the Caving Mining Method around the world.
Response to the reviewer
Thanks to the reviewer for constructive insight. In the introduction, we provide more information about Caving Mining Method worldwide. Hopefully, the cited references would improve the section more sufficient for the literature review type of research.
In the caving mining method, mining is held by breaking most or all the ore body. Caving mines are classified as Longwall, Sublevel Caving, and Block Caving by requirement factors, including ore strength, rock strength, deposit shape, deposit dip, deposit size, ore grade, ore uniformity, and depth [9]. Longwall is recommended for any ore strength, weak/moderate rock strength, tabular deposit shape, low/flat deposit dip, thin/wide deposit size, moderate ore grade, uniform grade, and moderate/deep depth [9]. Due to coal seams having a relatively flat dip, so most of the longwall is applied in coal mining. Longwall mining in coal seams is an underground mining technique where a tabular block 9longwall panel) of coal with a typical length of 1.5-3.0 km, a typical width of 200-300 m, and a typical height of 3.0-4.5 m is extracted. Two pairs of roadways are first driven outside the panel within the seam for access. Machines used for operations are to shear as coal cutter, belt conveyor for hauling, and hydraulic-powered roof supports provides temporary support during coal cutting [143]. Longwall mining applied intensively in Australia, China, and America, and a number in Ukraine, India, Turkey, Bangladesh, and Poland, [137-152]. In Indonesia, the longwall mining method is applied in Kutai Kertanegara, East Kalimantan, at the mining concession of PT Gerbang Daya Mandiri (GDM). GDM recoverable sub-bituminous coal reserves are approximately 29.2 million tons and have been planned for 1 million tons of annual production [153-157]. The second caving method, Sublevel Caving, is an underground mining method proposed for moderate and strong strength of ore, weak rock strength, tabular/massive deposit shape, steep deposit dip, large thick deposit size, moderate ore grade, moderate in uniformity grade, and moderate depth [9]. Sublevel caving is a mass mining method in which the ore is drilled and blasted while the waste rock caves and fills the space created by the extraction of ore. The ore body is divided into vertical intervals called sublevel intervals. The ore within each sublevel interval is drilled in a fan-shaped design at a constant horizontal distance along the production drift. Load Haul Dump machine load muck pile from the draw point [163]. Dilution becomes the issue in this method due to the strength of the ore body and rock mass. Sublevel caving is applied in iron mines in Ukraine, iron mines in Sweden, iron oxide mines in Norway, coal mines in Spain, coal mining in India, and gold mines in Australia [156-166]. The last caving method, block caving, is recommended for moderate and weak ore and rock strength, weak rock strength, tabular/thick deposit shape, steep deposit dip, very thick deposit size, moderate ore grade, moderate in uniformity grade, and moderate depth [9].
Reviewer
Point 6. You are kindly asked to indicate in your paper that caving method is also used in Ukraine Chile and Kazakhstan as described in references below
Shavarskyi, Ia., Falshtynskyi, V., Dychkovskyi, R., Akimov, O., Sala, D., & Buketov, V. (2022). Management of the longwall face advance on the stress-strain state of rock mass. Mining of Mineral Deposits, 16(3), 78-85.
Orellana, L. F., Castro, R., Hekmat, A., & Arancibia, E. (2017). Productivity of a continuous mining system for block caving mines. Rock Mechanics and Rock Engineering, 50(3), 657-663.
Matayev, A., Kainazarova, A., Arystan, I., Abeuov, Y., Kainazarov, A., Baizbayev, M., Demin, V., & Sultanov, M. (2021). Research into rock mass geomechanical situation in the zone of stope operations influence at the 10th Anniversary of Kazakhstan’s Independence mine. Mining of Mineral Deposits, 15(1), 103-111.
Response to the reviewer
Thanks to the reviewer for providing the references that completed our resume data. Block Caving mine in Chile (El Teniente) had included in our first manuscript, while the Lvivvuhillia SE Mine (Ukraine) and the 10th Anniversary of Kazakhstan’s Independence Mine (Kazakhstan) had not included. In this revised version, we synthesized the data from these papers and other references related to those mines and wrote them down in Appendix Table A1.
Reviewer
Point 7. It will be great if the authors show some description in context – Why it is important to conduct this study?
Response to the reviewer
We add the opinion at the end of the introduction as below.
This study is important in dissemination for practitioners, engineers, and academics about the block caving method, which is the most efficient method of modern and future mining for huge ore bodies. Studies in Indonesia are necessary to provide recommendations for the potential application of this mining method in utilizing the country's natural resources (especially metal deposits).
Reviewer
Point 8. The novelty of the paper must be highlighted in the conclusions section.
Response to the reviewer
The new requirements of ore body suitable mined using the transformed block caving method has been identified successively. This method can be applied on the blocky ore body with a thickness of 200-800 meters, various rock mass strengths until 300 MPa, from low to high (from 0.3% Cu until more than 1.0% Cu) but uniform grade, and a depth from 500 to 2200 meters. We also synthesize technical specifications for running block caving mines, including preparation methods, undercutting strategy, mine design, mining equipment, and monitoring systems. The potential ore bodies mined using the transformed block caving method have been recommended. The block caving method is promising for underground mining metal deposits in Indonesia, especially Cu-Au porphyry deposits. Although it is favorable for rock mass with various strengths and ore bodies with any grade level, the large dimension of the ore body is the fixed requirement because it needs fast and continuous production to comply with the specific pay-back period and settle the investment costs. So, porphyry is the appropriate ore body type to be mined using this method. Considering the requirements and the successful practice of block caving project in Grasberg Caving Complex as a role model, we suggest the Indonesian government, assisted by national or global investors, should concentrate on the detailed exploration of porphyry deposits and the feasibility studies applying the method on the prospective ore bodies i.e., Onto, Tambulilato, Tumpangpitu, and Randu Kuning. In addition, exploration method, cost, operational, mining policy, and social geology are important aspects worth noting.
Reviewer
Point 9. Please consider the suggested research in your paper when compared underground mining methods:
1). Ibishi, G., Yavuz, M., & Genis, M. (2020). Underground mining method assessment using decision-making techniques in a fuzzy environment: case study, Trepça mine, Kosovo. Mining of Mineral Deposits, 14(3), 134-140. https://doi.org/10.33271/mining14.03.134
Important issue: According to UBC approach six alternatives – Block Caving, Cut-and-Fill Stoping, Sub-level Caving, Sub-level Stoping, Square Set Stoping, and Top Slicing have been considered as technically feasible alternatives. Results shows that cutand-fill stoping method is the optimal mining method for deep excavation mining levels.
2). Wu, H., Yin, Z., Zhang, Y., Qi, C., Liu, X., & Wang, J. (2022). Optimization of underground coal mining methods based on life cycle assessment. Frontiers in Earth Science, 933. https://doi.org/10.3389/feart.2022.879082
Important issue: A summary comparison of the single scores of different mining methods suggests that the environmental burden of pillarless mining is the lowest, and as the width of the coal pillar gradually increases, its single score shows a trend of increasing and then decreasing. Therefore, the single score of non-pillar mining is the lowest compared to that of other mining methods and can be the optimal mining method.
Response to the reviewer
We studied this two recommended research, then considered them for mining method selection by writing in the chapter introduction.
Most of the highly productive mines in the world are excavated on the surface. Surface mining is more profitable as it has exceptional advantages in flexibility and mobilizing. So, realization design for producing ore as much as possible is convenient. Unfortunately, surface mining has an economic limit when the remaining ore reserve is getting deeper. In that case, underground mining methods are the only option. In addition, underground mining is assumed to leave fewer environmental impacts than surface mining [2]. Among underground mining methods, block caving is the most cost-effective as it can produce 10,000 – 100,000 tons per day with a relative operating cost of USD 1.0 to 2.5 per ton [3,4]. In the last twenty years, the existing block caving mines have varying ore body thickness ranges from 200 to 800 meters. A study proposes that cut and fills stopping method was selected as the optimal underground mining method for deep mining (>800 meters below the ground surface), compared with block caving and 4 other methods [131]. However, it is relevant for the lead-zinc-silver Trepca mineralization deposit investigated in the study, which has an irregular shape and ore thickness of 30-100 meters. Other studies on underground coal mining noted that non-pillar mining or Longwall Mining -a caving method for coal deposits- had the lowest environmental burden and was determined to be the optimal mining method [132]. Thus, the caving method can be favored as an underground mining method due to its high productivity and low environmental impact. The deposit size must be huge enough to settle the investment costs at the beginning of production.
Reviewer
Point 10. In general, the presented article leaves a positive impression and, after eliminating these comments and taking into account the recommendations made, it can be recommended for publication in the journal "Energies".
Response to the reviewer
We are supported by valuable insight and constructive recommendations. The discussion and detailed explanations added have enriched the knowledge contained in this paper. After accommodating all comments, we hope it becomes a reliable manuscript for publication in the journal.

Reviewer 3 Report
Exploitation of ore deposits with the use of the block caving method is particularly important in the case of thick ore deposits for which the parameters of the volume of production and exploitation losses are important. The presented list of mining projects and the possibilities of using the block caving method in Indonesia shows great opportunities to ensure the continuity of operation and directly translates into the economic growth of the country. Below are some comments and suggestions:
1. In the introduction, a few sentences should be added regarding the thickness of the exploited thick deposits using the block caving method.
2. In the second chapter, line 172: it should be written what means: "The grade of ore is should not high". It should be written down what percentage range is being considered (indicative).
3. Line 298, subsection should rather have number 3. Moreover, it should be written in one sentence what is the depth range of the exploited deposits (from to).
4. In the fourth chapter, some information on the technology of exploitation should be added: The length of the blast holes drilled in the roof could be mentioned; quantity of explosives per hole; number of holes in the fan; machines used for haulage of the output, type of support and what is the daily output.
5. With reference to the sixth chapter, it should be stated whether remotely controlled mining machines are taken into account.
6. In the conclusions, it should be stated how many years the concession for currently exploited fields covers (to which year the production is planned).
Author Response
Reviewer 3
Reviewer
Exploitation of ore deposits with the use of the block caving method is particularly important in the case of thick ore deposits for which the parameters of the volume of production and exploitation losses are important. The presented list of mining projects and the possibilities of using the block caving method in Indonesia shows great opportunities to ensure the continuity of operation and directly translates into the economic growth of the country. Below are some comments and suggestions:
- In the introduction, a few sentences should be added regarding the thickness of the exploited thick deposits using the block caving method.
Response to reviewer
We thank the reviewer for the constructive comment and questions. The reviewer's purpose/message for this manuscript has been well understood and received. We have updated this latest manuscript per the reviewers' recommendations by adding some technical explanations and more precise conclusions.
- In the introduction (Section 1), we added the following information.
The thickness of the ore body mined using the block caving method varies between 200 to 800 meters. As an indication, the ore body widths of El-Teniente, Grasberg, Northparkes, Oyu Tulgui, Palabora, Ridgeway, and Stornoway are 500-800, 400, 200, 500, 250, 200, and 225, respectively.
Reviewer
- In the second chapter, line 172: it should be written what means: "The grade of ore is should not high". It should be written down what percentage range is being considered (indicative).
Response to reviewer
- We add the following description to explain the sentence: "The grade of ore is should not high"
The grade of ore is should not high means (point 6) that the uniformity of grades in the ore body is more important than a certain percentage of grades because in this method almost all parts of deposits can be recovered. The consideration of the mine feasibility depends more on the volume of ore – also the other contained minerals- that will determine the life of the mine. The high-grade Cu deposits containing low-grade other minerals is equivalent with the low-grade Cu deposits containing high-grade other minerals. As quantitative values for comparison, Ridgeway Deeps has 0.38% Cu and 1.80 g/t Au [83]. While El-Teniente ore body has 0.62-0.98% Cu, 0.019% Mo, 0.005 g/t Au, and 0.5 g/t Ag [4,94,136]. Both are mined economically using block caving method.
Reviewer
- Line 298, subsection should rather have number 3. Moreover, it should be written in one sentence what is the depth range of the exploited deposits (from to).
Response to reviewer
3.We replace subsection 3.3. with section 3. We add an explanation about the depth range in number 4 as below.
On the latest block caving mining projects, orebody lies at depths from 500 up to 2,200 m below the surface.
Reviewer
- In the fourth chapter, some information on the technology of exploitation should be added: The length of the blast holes drilled in the roof could be mentioned; quantity of explosives per hole; number of holes in the fan; machines used for haulage of the output, type of support and what is the daily output.
Response to reviewer
- As per our review of documented sources, the detailed technical note discussing this operational practice of blasting, hauling, and supporting were insufficient. Nonetheless, we tried to write a summary and approach from the available references as follows:
No detailed technical notes related to blasting practice specifications at Grasberg Caving Complex were documented. Based on the underground ring blasting design (long hole blasting techniques applied in block caving), the most common diameter range ranges from 64 to 115 mm. For draw belling, a fan of 102 mm diameter consists of 9 holes with the length of blast hole drilled in the roof range from 8 to 22.9 m. By burdening 2.5 – 2.6 m, it consumes about 58 to 188 kg emulsion (1.1 gr/cc) per hole: the smaller hole diameter, 89 mm, is used on a narrow inclined undercut ring to avoid dilution. The burden is designed to be closer, only ranging from 1.8 to 2.1 m, with 5 holes in a fan. It needs from 8.7 to 69 kg emulsion (1.0 gr/cc) per hole [130]. As the draw belling and undercutting blasting output, the broken ore was mucked from the draw point using Load Haul Dump (LHDs) operated remotely, the delivered via loading chute to a rail haulage system. Wire mesh and shotcrete were applied to support the draw point and undercut level. Blasting conducts for 2 rings (fan series) every day and results in about 80-100 kilotons the Cu-Au fragmented ore.
Reviewer
- With reference to the sixth chapter, it should be stated whether remotely controlled mining machines are taken into account.
Response to reviewer
- We add a discussion on the mining machine using a remote control system in sub-section 6.3. Operational.
6.3. Operational
Block caving is a mining method with large productivity/mass mining. Remote-control systems for ore drawing and mucking are unavoidable to improve safety and pursue production targets. The deployment and mastery of this remote-control technology system should be supported by providing training/internships for prospective operators working in block caving mines. Mastery of the technology of controlling remote mining will be beneficial for the availability of Indonesian human resources who are competent in operating block caving mining systems. The possibility of adopting a block caving or even the raise caving method should be considered for optimum mining practices.
Reviewer
- In the conclusions, it should be stated how many years the concession for currently exploited fields covers (to which year the production is planned).
Response to reviewer
- In the conclusion, we write the explanation related to the estimation time of mining stages for the recommendation of concession.
The development stage of block caving infrastructure needs 3-5 years until ready for continuous production. The existing block caving projects have a life of mine for production, ranging from 12 to 35 years. Thus, we propose a long-term concession, about 15 until 40 years, covering development and operational production stages.

Round 2
Reviewer 2 Report
Dear authors,
I am more than satisfied with the corrections provided by you.
This study is an important contribution to sustainable mining.
Congratulations to the authors.
Author Response
Dear reviewer,
We are glad to read that you have been satisfied with the revised manuscripts.
Hopefully this paper can enrich the reference to mining sustainability.
Thank you very much.